# On Efficient Estimation of Distributional Treatment Effects under Covariate-Adaptive Randomization

**Undral Byambadalai** [1]   **Tomu Hirata** [2]   **Tatsushi Oka** [3]   **Shota Yasui** [1]

## Abstract

This paper focuses on the estimation of distributional treatment effects in randomized experiments that use covariate-adaptive randomization (CAR). These include designs such as Efron's biased-coin design and stratified block randomization, where participants are first grouped into strata based on baseline covariates and assigned treatments within each stratum to ensure balance across groups. In practice, datasets often contain additional covariates beyond the strata indicators. We propose a flexible distribution regression framework that leverages off-the-shelf machine learning methods to incorporate these additional covariates, enhancing the precision of distributional treatment effect estimates. We establish the asymptotic distribution of the proposed estimator and introduce a valid inference procedure. Furthermore, we derive the semiparametric efficiency bound for distributional treatment effects under CAR and demonstrate that our regression-adjusted estimator attains this bound. Simulation studies and empirical analyses of microcredit programs highlight the practical advantages of our method.

## 1. Introduction

Randomized experiments have been fundamental in uncovering the impact of interventions and shaping policy decisions since the seminal work by Fisher (1935). The use of randomized experiments to estimate causal effects has been extensively adopted across diverse scientific fields (Rubin, 1974; Heckman et al., 1997; Imai, 2005; Imbens & Rubin, 2015) and has also become a widely accepted practice in the technology industry (Tang et al., 2010; Bakshy et al., 2014; Kohavi et al., 2020).

In this paper, we examine the estimation of distributional treatment effects in randomized experiments that employ covariate-adaptive randomization (CAR). Under CAR, individuals are first partitioned into *strata* based on similar covariates, and treatments are then assigned within each stratum to ensure balance across groups. The CAR framework encompasses various randomization schemes, from stratified block randomization to Efron's biased coin design (Imbens & Rubin, 2015), with simple random sampling as a special case.

We introduce a regression adjustment method for estimating distributional treatment effects under CAR when auxiliary data beyond stratum indicators are available. Incorporating this data enhances estimation precision. While experimental analyses often focus on the average treatment effect (ATE), relying solely on the ATE can overlook important insights. Examining distributional treatment effects provides a more comprehensive understanding of the treatment impact by capturing changes in the entire outcome distribution.

Our approach employs a distribution regression framework, leveraging the Neyman-orthogonal moment condition (Chernozhukov et al., 2018) to ensure first-order insensitivity to nuisance parameter estimation. These nuisance parameters—conditional outcome distributions given pre-treatment covariates—are estimated using machine learning techniques such as random forests, neural networks, and gradient boosting, enabling flexibility in handling complex and high-dimensional data. Incorporating cross-fitting further strengthens robustness against estimation errors.

Randomization schemes under CAR are extensively used across various disciplines. In clinical trials, stratified block randomization ensures balanced treatment allocation across key covariates such as age, gender, and disease severity (Rosenberger & Lachin, 2015). In social experiments, researchers often stratify by geographical regions or socioeconomic characteristics (Duflo et al., 2007; Bruhn & McKenzie, 2009), while in the technology sector, covariate-adaptive designs enhance precision in A/B testing (Xie & Aurisset, 2016).

[1] CyberAgent, Inc., Tokyo, Japan [2] Databricks Japan, Inc., Tokyo, Japan [3] Department of Economics, Keio University, Tokyo, Japan. Correspondence to: Undral Byambadalai <undral_byambadalai@cyberagent.co.jp>.

*Proceedings of the 42nd International Conference on Machine Learning*, Vancouver, Canada. PMLR 267, 2025. Copyright 2025 by the author(s).

Our paper makes several key contributions. First, we extend the applicability of regression adjustment under CAR to estimate distributional treatment effects. While regression adjustment is widely used to reduce variance in ATE estimation under simple random sampling (Freedman, 2008a; Lin, 2013) and has recently been studied in the context of CAR (Rafi, 2023; Cytrynbaum, 2024; Wang et al., 2023), our work advances this framework to accommodate distributional treatment effects. Unlike Jiang et al. (2023), who focus on regression adjustment for quantile treatment effects and assume continuous outcomes, our method is applicable to both discrete and mixed discrete-continuous outcomes.

Second, we establish the limit distribution of our estimator within the asymptotic framework for CAR, extending beyond the standard i.i.d. treatment assignment structure in causal inference. Third, we derive the semiparametric efficiency bound for distributional treatment effects under CAR and demonstrate that our estimator attains this bound. Finally, through simulations and an empirical analysis of microcredit programs, we illustrate the effectiveness of our approach in practical settings.

The rest of the paper is organized as follows. Section 2 reviews the literature, and Section 3 provides the setup. Section 4 introduces distributional treatment effect parameters, their identification, and estimation. Section 5 presents asymptotic results. Section 6 discusses findings from simulated and real data. Section 7 concludes. Appendix contains notations, proofs, and additional experimental results.

## 2. Related Literature

**Distributional Treatment Effects** Distributional and quantile treatment effects have long been recognized as important parameters to estimate beyond the mean effects. The quantile treatment effect was first introduced by Doksum (1974) and Lehmann & D'Abrera (1975). Subsequently, estimation and inference methods for distributional and quantile treatment effects have been developed and applied in econometrics, statistics and machine learning community, including Heckman et al. (1997); Imbens & Rubin (1997); Abadie (2002); Abadie et al. (2002); Chernozhukov & Hansen (2005); Koenker (2005); Bitler et al. (2006); Athey & Imbens (2006); Firpo (2007); Chernozhukov et al. (2013); Koenker et al. (2017); Callaway et al. (2018); Callaway & Li (2019); Chernozhukov et al. (2019); Ge et al. (2020); Zhou et al. (2022); Park et al. (2021); Kallus & Oprescu (2023); Oka & Yamada (2023); Näf & Susmann (2024); Xu et al. (2025), among others. Most of this work explore the conditional distributional and quantile treatment effects. Oka et al. (2024) and Byambadalai et al. (2024) consider the estimation of unconditional distributional treatment effects but under simple random sampling. Kallus et al. (2024) address problems in which nuisance parame-

ters depend on the target parameter itself, as seen in cases like the quantile treatment effect (QTE) and local QTE. In contrast, our estimation of the distributional treatment effect involves nuisance parameters that correspond to conditional means, which can be effectively estimated using machine learning algorithms.

**Conditional Average Treatment Effects** An alternative method for examining heterogeneity in treatment effects is to condition on observed variables and estimate the Conditional Average Treatment Effect (CATE) (Imai & Ratkovic, 2013; Athey & Imbens, 2016; Johansson et al., 2016; Shalit et al., 2017; Alaa & Van Der Schaar, 2017; Wager & Athey, 2018; Chernozhukov et al., 2018; Künzel et al., 2019; Shi et al., 2019; Nie & Wager, 2021; Guo et al., 2023; Sverdrup & Cui, 2023; Van Der Laan et al., 2023). CATE quantifies the ATE within subgroups defined by observed attributes, such as gender, age, or prior platform use, thereby capturing the heterogeneity that can be explained by observable information. In contrast, our approach is designed to measure unobserved heterogeneity and can be extended to estimate distributional parameters conditional on observed data.

**Regression adjustment under covariate-adaptive randomization** The literature on utilizing pre-treatment covariates to reduce variance in estimating the ATE under simple random sampling is extensive, beginning with Fisher (1932) and followed by contributions from Cochran (1977); Yang & Tsiatis (2001); Rosenbaum (2002); Freedman (2008a;b); Tsiatis et al. (2008); Rosenblum & Van Der Laan (2010); Lin (2013); Berk et al. (2013); Ding et al. (2019), among others.

In the context of covariate-adaptive randomization, some recent studies have explored regression adjustment for estimating the ATE. Recent work by Cytrynbaum (2024) derives an asymptotically optimal linear covariate adjustment tailored to a given stratification. Similarly, Rafi (2023) investigates regression adjustment and establishes the semiparametric efficiency bound for estimating the ATE under covariate-adaptive randomization. Bai et al. (2024) examines covariate adjustment within a "matched pairs" design, where each stratum consists of two observations, with one randomly assigned to treatment. In biostatistics, Bannick et al. (2023) and Tu et al. (2023) analyze general approaches to covariate adjustment under covariate-adaptive randomization, while Wang et al. (2023) considers parameters defined by estimating equations. Notably, most of these studies concentrate on ATE estimation, with the exception of Jiang et al. (2023), who investigates the estimation of quantile treatment effects in the same setting.

**Semiparametric Estimation** Our work builds on the semiparametric estimation literature, which addresses the challenge of estimating low-dimensional parameters in the presence of high-dimensional nuisance parameters. This

**Illustration: Comparison of Randomization Methods**

| Sample with Strata | Simple Random Sampling | Stratified Block Randomization |
|---|---|---|

*Figure 1.* An illustration of treatment and control assignments under simple random sampling (SRS) and stratified block randomization (SBR). While both allocate 50 subjects per group, SRS may result in imbalanced group compositions, whereas SBR preserves the proportional representation of strata in each group, matching the overall sample distribution.

literature includes fundamental contributions from Robinson (1988); Bickel et al. (1993); Newey (1994); Robins & Rotnitzky (1995), and more recent developments by Chernozhukov et al. (2018); Ichimura & Newey (2022). Our setup can be characterized as a semiparametric problem with Neyman-orthogonal moment conditions, as outlined in Neyman (1959); Chernozhukov et al. (2022).

## 3. Setup and Notation

We consider randomized controlled trials (RCT) with covariate-adaptive randomization when there are multiple treatment arms. In covariate-adaptive randomization, individuals are first grouped into *strata* based on similar values of their baseline covariates. Within each stratum, treatments are assigned to achieve "balance" across groups, typically using complete randomization within each stratum. This approach ensures that the treatment allocation accounts for covariate distributions within strata, thereby improving comparability between treatment groups.

Figure 1 provides a simple illustration. Consider an experiment involving 100 subjects, where 50% are assigned to a treatment group and 50% to a control group. Suppose the subjects are divided into two strata based on baseline covariates (e.g., Stratum 1 represents younger individuals, and Stratum 2 represents older individuals). Stratum 1 comprises 30 subjects, while Stratum 2 comprises 70. Under simple random sampling (SRS), subjects are randomly assigned to treatment or control groups without regard to strata. In contrast, stratified block randomization (SBR) independently assigns subjects within each stratum, ensuring proportional representation. While both methods allocate exactly 50 subjects to each group, SRS does not guarantee that the composition of treatment groups reflects the overall sample distribution. For example, Figure 1 depicts one possible outcome, where Stratum 2 (older individuals)

is over-represented in the treatment group under SRS. By contrast, SBR achieves balance within each stratum and maintains the same composition in treatment and control groups as in the full sample.

Let $Y_i \in \mathcal{Y} \subset \mathbb{R}$ denote the observed outcome of interest for the $i$th unit and $W_i \in \mathcal{W} := \{1, \ldots, |\mathcal{W}|\}$ denote the index of the treatment received by the $i$th unit for each unit $i \in [n] := \{1, \ldots, n\}$, where $n$ denotes the sample size. Also, let $S_i \in \mathcal{S} := \{1, ..., S\}$ denote the stratum variable and $X_i \in \mathcal{X} \subset \mathbb{R}^{d_x}$ denote the extra covariates besides $S_i$. We allow $X_i$ and $S_i$ to be dependent. To rule out empty stratum, we assume the probability of individuals being assigned to each stratum is positive, i.e., $p(s) := \mathbb{P}(S_i = s) > 0$ for every $s \in \mathcal{S}$. We adapt the potential outcome framework (Rubin, 1974; Imbens & Rubin, 2015), and let $Y_i(w)$ denote the potential outcome under treatment $w \in \mathcal{W}$ for the $i$th unit. The observed outcome and potential outcomes are related to treatment assignment by the relationship $Y_i = Y_i(W_i)$.

In order to describe the treatment assignment mechanism, we let $\pi_w(s) := \mathbb{P}(W_i = w | S_i = s) \in (0, 1)$ be the target assignment probability for treatment $w$ in stratum $s$, which may vary across strata. We define sub-sample as $n_w(s) := \sum_{i=1}^n \mathbb{1}\{W_i = w, S_i = s\}$ and $n(s) := \sum_{i=1}^n \mathbb{1}\{S_i = s\}$, where $\mathbb{1}\{\cdot\}$ denotes the indicator function. The empirical analogues of $p(s)$ and $\pi_w(s)$ are given by $\widehat{p}(s) := n(s)/n$ and $\widehat{\pi}_w(s) := n_w(s)/n(s)$, respectively. In line with Bugni et al. (2019), we impose the following assumptions on the treatment assignment mechanism.

**Assumption 3.1.** We have

(i) $\{Y_i(1), \ldots, Y_i(|\mathcal{W}|), S_i, X_i\}_{i=1}^n$ are i.i.d.

(ii) $\{Y_i(1), \ldots, Y_i(|\mathcal{W}|), X_i\}_{i=1}^n \perp\!\!\!\perp \{W_i\}_{i=1}^n | \{S_i\}_{i=1}^n$.

(iii) $\widehat{\pi}_w(s) = \pi_w(s) + o_p(1)$ for every $(w, s) \in \mathcal{W} \times \mathcal{S}$.

Assumption 3.1 (i) allows for cross-sectional dependence among treatment statuses $\{W_i\}_{i=1}^n$, thereby accomodating many covariate-adaptive randomization schemes. Assumption 3.1 (ii) states that the assignment is independent of potential outcomes and pre-treatment covariates conditional on strata. Assumption 3.1 (iii) states the assignment probabilities converge to the target assignment probabilities as sample size increases.

Common randomization schemes satisfying Assumption 3.1 include simple random sampling, stratified block randomization, biased-coin design (Efron, 1971), and adaptive biased-coin design (Wei, 1978). In Section 6.2, we reanalyze a field experiment on microcredit programs, where stratification occurs at the provincial level, and treatments are randomly assigned within each province based on target probabilities.

# 4. Distributional Treatment Effects

The parameters of interest are based on the distribution functions of potential outcomes, denoted by

$$F_{Y(w)}(y) := \mathbb{P}\big(Y(w) \le y\big),$$

for $y \in \mathcal{Y}$ and $w \in \mathcal{W}$.

First, we define the *distributional treatment effect* (DTE) between treatments $w, w' \in \mathcal{W}$ as

$$\Delta_{w,w'}^{DTE}(y) := F_{Y(w)}(y) - F_{Y(w')}(y),$$

for $y \in \mathcal{Y}$. The DTE measures the difference between the distribution functions of the potential outcomes.

We also define the *probability treatment effect* (PTE) between treatments $w, w' \in \mathcal{W}$ as

$$\Delta_{w,w'}^{PTE}(y_j) := f_{Y(w)}(y_j) - f_{Y(w')}(y_j),$$

for each $j = 1, \ldots, J$, where, given a set of points $\mathcal{Y}_J := \{y_1, \cdots, y_J\} \subset \mathcal{Y}$ with $y_0 = -\infty$, the probability mass function $f_{Y(w)}(\cdot)$ is defined as

$$\begin{aligned} f_{Y(w)}(y_j) :&= \mathbb{P}\big(y_{j-1} < Y(w) \le y_j\big) \\ &= F_{Y(w)}(y_j) - F_{Y(w)}(y_{j-1}). \end{aligned}$$

Each probability $f_{Y(w)}(y_j)$, which we refer to as the *bin probability*, can be obtained as a difference between the values of the distribution function at $y_j$ and $y_{j-1}$. The PTE quantifies the difference in probabilities across bins, effectively capturing differences in "histograms" of the potential outcomes.

## 4.1. Identification

Although the potential outcomes $\{Y(w) : w \in \mathcal{W}\}$ are unobserved variables, the conditional distribution functions of these outcomes, $F_{Y(w)}(\cdot|S = s)$, can be identified. This is because, under Assumption 3.1, $F_{Y(w)}(\cdot|S = s)$ coincides with $F_Y(\cdot|W = w, S = s)$, the conditional distribution function of the observed outcome given treatment $w$ within stratum $s$. By the law of total probability, the distribution function $F_{Y(w)}(y)$ is then expressed as, for any $y \in \mathcal{Y}$,

$$F_{Y(w)}(y) = \sum_{s=1}^{S} p(s) F_{Y(w)}(y|S = s). \tag{1}$$

Hence, $F_{Y(w)}(y)$ is identifiable under Assumption 3.1.

## 4.2. Empirical estimator

Under the RCT setting, the distribution function can be calculated without conditioning on pre-treatment covariates, unlike observational studies. Under CAR, we define an empirical estimator of the distribution function $F_{Y(w)}(y)$ for $y \in \mathcal{Y}$ and treatment $w \in \mathcal{W}$ as:

$$\widehat{F}_{Y(w)}^{emp}(y) := \frac{1}{n} \sum_{i=1}^{n} \frac{\mathbb{1}\{W_i = w\} \cdot \mathbb{1}\{Y_i \le y\}}{\widehat{\pi}_w(S_i)}. \tag{2}$$

This estimator aggregates the empirical distribution functions across strata, and takes the form of an inverse-propensity weighting (IPW) estimator Rosenbaum & Rubin (1984). The empirical estimator for the DTE is then formed as

$$\widehat{\Delta}_{w,w'}^{DTE,emp}(y) := \widehat{F}_{Y(w)}^{emp}(y) - \widehat{F}_{Y(w')}^{emp}(y).$$

While the estimator is unbiased and consistent, its efficiency can be enhanced by leveraging pre-treatment covariates.

## 4.3. Regression-adjusted estimator

To incorporate pre-treatment covariates $X$, we adopt the distribution regression framework, treating the conditional distribution function $\mu_w(y, s, x) := F_{Y(w)}(y|S = s, X = x)$ as the mean regression for a binary outcome $\mathbb{1}\{Y(w) \le y\}$. Specifically, for each $y \in \mathcal{Y}$ and $w \in \mathcal{W}$, we can write

$$\mu_w(y, S, X) = \mathbb{E}[\mathbb{1}\{Y(w) \le y\}|S, X].$$

The conditional mean function can be estimated at each location $y \in \mathcal{Y}$ using supervised learning algorithms, such as LASSO, random forests, boosted trees, or deep neural networks. Let $\widehat{\mu}_w(\cdot)$ be an estimator for $\mu_w(\cdot)$.

The regression-adjusted estimator of $F_{Y(w)}(y)$ for each $w \in \mathcal{W}$ and $y \in \mathcal{Y}$ is then defined as

$$\widehat{F}_{Y(w)}^{adj}(y) = \frac{1}{n} \sum_{i=1}^{n} \Psi_i(y), \tag{3}$$

where

$$\Psi_i(y) := \frac{\mathbb{1}\{W_i = w\} \cdot \left( \mathbb{1}\{Y_i \le y\} - \widehat{\mu}_w(y, S_i, X_i) \right)}{\widehat{\pi}_w(S_i)}$$
$$+ \widehat{\mu}_w(y, S_i, X_i).$$

The regression-adjusted estimator for DTE can then be formed as

$$\widehat{\Delta}_{w,w'}^{DTE,adj}(y) := \widehat{F}_{Y(w)}^{adj}(y) - \widehat{F}_{Y(w')}^{adj}(y). \qquad (4)$$

The estimator takes the form of the well-known augmented inverse-propensity weighting (AIPW) estimator, relying on a doubly-robust moment condition (Robins et al., 1994; Robins & Rotnitzky, 1995). Specifically, the moment condition exhibits the Neyman orthogonality property (Chernozhukov et al., 2018; 2022), making our estimator first-order insensitive to estimation errors in the nuisance parameters $\mu_w(\cdot)$. To further improve robustness, we apply cross-fitting as outlined in Chernozhukov et al. (2018). The estimation procedure is outlined in Algorithm 1.

The empirical and adjusted estimators for the PTE can be defined in a similar fashion. See Appendix B for the details.

---

**Algorithm 1** ML regression-adjusted DTE estimator with cross-fitting

---

**Input:** Data $\{(Y_i, W_i, X_i, S_i)\}_{i=1}^n$ split randomly into $L$ roughly equal-sized folds ($L > 1$); $\mathcal{M}$ a supervised learning algorithm
**for** level $y \in \mathcal{Y}$ **do**
    **for** (treatment $w \in \mathcal{W}$, stratum $s \in \mathcal{S}$, fold $\ell$=1 to$L$) **do**
        Train $\mathcal{M}$ on data excluding fold $\ell$, using observations in treatment group $w$ within stratum $s$.
        Use $\mathcal{M}$ to obtain predictions $\hat{\mu}_w(y, S_i, X_i)$ for all observations in stratum s for fold $\ell$.
    **end for**
    Compute $\widehat{F}_{Y(w)}^{adj}(y)$ according to Eq. (3).
    Compute $\widehat{\Delta}_{w,w'}^{DTE,adj}(y)$ using Eq. (4) for $w, w' \in \mathcal{W}$.
**end for**
**Output:** DTE estimator $\{\widehat{\Delta}_{w,w'}^{DTE,adj}(y)\}_{y \in \mathcal{Y}}$.

---

# 5. Asymptotic Distribution of the DTE estimator

In this section, we derive the asymptotic distribution of the proposed estimator, which enables statistical inference and the construction of confidence intervals. Additionally, we establish the semiparametric efficiency bound for the DTE and demonstrate that our estimator achieves this bound under the specified assumptions. We begin by introducing some additional notation to formalize our results. Let $\| \cdot \|_{P,q}$ denote the $L_q(P)$ norm, and $L^\infty(\mathcal{Y})$ be the space of

uniformly bounded functions mapping an arbitrary index set $\mathcal{Y}$ to the real line.

**Assumption 5.1.** (i) For $w \in \mathcal{W}$, $\delta_w(y, s, X_i) := \widehat{\mu}_w(y, s, X_i) - \mu_w(y, s, X_i)$, we have

$$\sup_{y \in \mathcal{Y}, s \in \mathcal{S}} \left| \frac{\sum_{i \in I_w(s)} \delta_w(y, s, X_i)}{n_w(s)} - \frac{\sum_{i \in I_{w'}(s)} \delta_w(y, s, X_i)}{n_{w'}(s)} \right|$$
$$= o_p(n^{-1/2}),$$

where $I_w(s) := \{i \in [n] : W_i = w, S_i = s\}$.

(ii) For $w \in \mathcal{W}$, let $\mathcal{F}_w = \{\mu_w(y, s, x) : y \in \mathcal{Y}\}$ with an envelope $F_w(s, x)$. Then, $\max_{s \in \mathcal{S}} \mathbb{E}[|F_w(S_i, X_i)|^q | S_i = s] < \infty$ for $q > 2$ and there exist fixed constants $(\alpha, v) > 0$ such that

$$\sup_Q N\left(\varepsilon \|F_w\|_{Q,2}, \mathcal{F}_w, L_2(Q)\right) \le \left(\frac{\alpha}{\varepsilon}\right)^v, \quad \forall \varepsilon \in (0, 1],$$

where $N(\cdot)$ denotes the covering number and the supremum is taken over all finitely discrete probability measures $Q$.

Assumption 5.1(i) provides a high-level condition on the estimation of $\widehat{\mu}_w(y, s, X_i)$. Assumption 5.1(ii) imposes mild regularity conditions on $\mu_w(y, s, X_i)$. Specifically, Assumption 5.1(ii) holds automatically when $\mathcal{Y}$ is a finite set.

We now establish the weak convergence of our proposed estimator in the following theorem, which serves as the theoretical foundation for statistical inference. To that end, we define the following terms. Letting $\mu_w(y, s) := \mathbb{E}[\mu_w(y, S_i, X_i) | S_i = s]$, we define

$$\zeta_i(y) := \mu_w(y, S_i) - \mu_{w'}(y, S_i),$$
$$\phi_i(y, w) := \frac{\mathbb{1}\{Y_i(w) \le y\} - \mu_w(y, s)}{\pi_w(s)}$$
$$+ \left(1 - \frac{1}{\pi_w(s)}\right) (\mu_w(y, s, X_i) - \mu_w(y, s))$$
$$- (\mu_{w'}(y, s, X_i) - \mu_{w'}(y, s)).$$

**Theorem 5.2** (Asymptotic Distribution). *Suppose Assumption 3.1 and 5.1 hold. Then, for every $w, w' \in \mathcal{W}$, in $L^\infty(\mathcal{Y})$, uniformly over $y \in \mathcal{Y}$, the regression-adjusted estimator defined in Algorithm 1 satisfies*

$$\sqrt{n}(\widehat{\Delta}_{w,w'}^{DTE,adj}(y) - \Delta_{w,w'}^{DTE}(y)) \rightsquigarrow \mathcal{G}(y),$$

*where $\mathcal{G}(y)$ is a Gaussian process with covariance kernel $\Omega(y, y')$, which is given by*

$$\Omega(y, y') := \Omega_1(y, y', w) + \Omega_1(y, y', w') + \Omega_2(y, y'),$$

*with $\Omega_1(y, y', w) := \mathbb{E}[\pi_w(S_i)\phi_i(y, w)\phi_i(y', w)]$ and $\Omega_2(y, y') := \mathbb{E}[\xi_i(y)\xi_i(y')]$.*

We next derive the semiparametric efficiency bound and show our estimator achieves this bound in the following theorem. This implies that the asymptotic variance of any regular, consistent, and asymptotically normal estimator of DTE cannot be smaller than this variance.

**Theorem 5.3** (Semiparametric Efficiency Bound)**.**

(a) *Under Assumption 3.1, the semiparametric efficiency bound for $\Delta_{w,w'}^{DTE}(y)$ for a given $y \in \mathcal{Y}$ is $\Omega(y)$, which is defined by*

$$\Omega(y) := \Omega_1(y,y,w) + \Omega_1(y,y,w') + \Omega_2(y,y),$$

*where $\Omega_1(\cdot)$ and $\Omega_2(\cdot)$ are defined in Theorem 5.2.*

(b) *Under Assumptions 3.1 and 5.1, the regression-adjusted estimator $\widehat{\Delta}_{w,w'}^{DTE,adj}(y)$ attains the semiparametric efficiency bound.*

As a corollary to this theorem, it follows that the variance of the regression-adjusted estimator with known nuisance functions is smaller than that of the empirical estimator, since the latter can be regarded as a special case in which the adjustment terms $\widehat{\mu}_w(\cdot)$ are set to zero.

**Corollary 5.4** (Variance reduction)**.** *Under Assumption 3.1, for each $y \in \mathcal{Y}$,*

$$Var\big(\tilde{\Delta}_{w,w'}^{DTE,adj}(y)\big) \leq Var\big(\widehat{\Delta}_{w,w'}^{DTE,emp}(y)\big),$$

*where $\tilde{\Delta}_{w,w'}^{DTE,adj}(y)$ is the regression-adjusted DTE estimator that incorporates known adjustment terms.*

Theorem 5.3 and Corollary 5.4 indicate that, asymptotically, regression adjustment enhances the precision of the DTE estimates compared to the unadjusted empirical estimator. In the following section, we evaluate the finite sample performance of our estimators using both simulated and real datasets.

# 6. Experiments

## 6.1. Simulation Study

In this section, we examine the finite sample performance of the estimators through a simulation study. The design consists of four strata ($S = 4$) constructed by partitioning the support of $Z_i \sim U(0,1)$ into $S$ equal-length intervals, where $S_i$ indicates the interval containing $Z_i$. For each unit $i$, we draw a 20-dimensional covariate vector $X_i = (X_{1,i}, \ldots, X_{20,i})^\top$ from a multivariate normal distribution $\mathcal{N}(0, I_{20 \times 20})$. The treatment indicator $W_i$ follows a Bernoulli distribution with probability 0.5 within each stratum, maintaining a constant target proportion of treated units ($W_i = 1$) across strata with $\pi_w(s) = 0.5$ for all $s \in \mathcal{S}$. We generate the outcome variable $Y_i$ as follows:

$$Y_i = b(X_i) + c(X_i)W_i + \gamma Z_i + u_i,$$

where $b(X_i)$ is the scaled Friedman (1991) function given by $b(X_i) = \sin(\pi X_{i1} X_{i2}) + 2(X_{i3} - 0.5)^2 + X_{i4} + 0.5 X_{i5}$, the treatment effect function is $c(X_i) = 0.1(X_{i1} + \log(1 + \exp(X_{i2})))$ and $\gamma = 0.1$. The error term is $u_i \sim \mathcal{N}(0,1)$. This data generating process introduces a complex, highly nonlinear relationship between covariates and the outcome, while including many covariates that do not affect the outcome. The heterogeneous treatment effects are correlated with specific covariates. This setup is a modified version of the setups in Nie & Wager (2021) and Guo et al. (2021).

We draw a sample of size 5,000 from the data-generating process and estimate the DTE at quantiles $\{0.1, \ldots, 0.9\}$ using empirical and regression-adjusted estimators. To approximate the ground truth, a separate sample of size $10^6$ is drawn, and the DTE is computed at the same locations. Linear and machine learning (ML) regression adjustments are implemented using linear regression and gradient boosting, with 2-fold cross-fitting. Linear regression is included as a baseline for comparison, as it has traditionally been widely used for regression adjustment in estimating average treatment effects.

Figure 2 presents the results from 1,000 simulation runs with a sample size of $n = 1,000$. Both the root mean squared error (RMSE) and the length of the confidence intervals are smaller for the linear adjustment method, and even further reduced with ML adjustment, compared to the empirical estimator. The confidence intervals are calculated using sample estimates of the asymptotic variance. The empirical estimator achieves a 95% confidence interval coverage close to the nominal level of 0.95. In contrast, regression-adjusted estimators show slight over-coverage, with coverage rates ranging from approximately 0.96 to 0.97 for linear adjustment and 0.97 to 0.99 for ML adjustment. Additional results on RMSE reductions across quantiles for sample sizes $n \in \{1,000, 5,000, 10,000\}$ are provided in Appendix Section D.1.

## 6.2. Real Data: The Impacts of Microfinance

We reexamine the randomized field experiment conducted by Attanasio et al. (2015) in 2008 to evaluate the impact of a joint-liability microcredit program targeting women. The study took place in northern Mongolia, encompassing 40 villages across five provinces. Randomization occurred at the village level, with three treatment groups: joint-liability (group) lending, individual lending, and a control group. Stratification was carried out at the provincial level to ensure balance ($S_i \in \{1, \ldots, 5\}$), as complete randomization could have led to some provinces containing only some particular treatment or control villages. This type of geographic stratification is commonly employed in social sciences, medicine and other disciplines to facilitate robust comparisons between treatment groups.

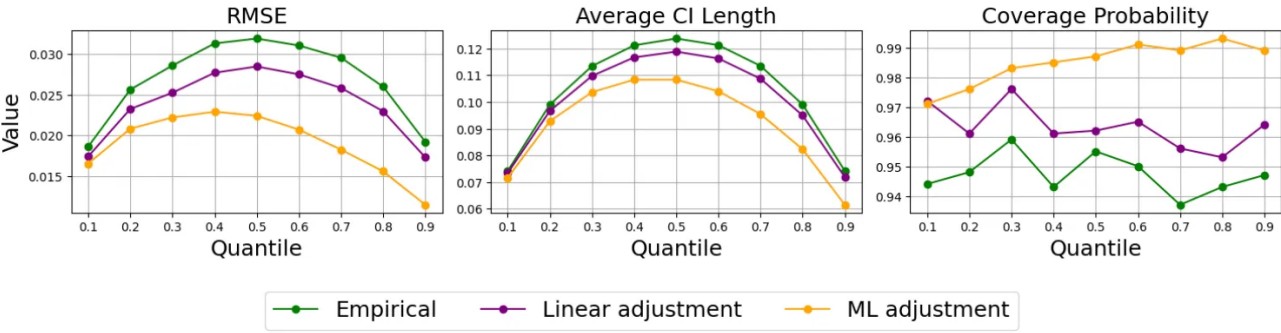

*Figure 2.* RMSE, average length and coverage probability of 95% confidence intervals (CI) on simulated data ($n = 1,000$). Linear adjustment uses linear regression, and machine learning (ML) adjustment uses gradient boosting, both with 2-fold cross-fitting. Number of simulations is 1,000.

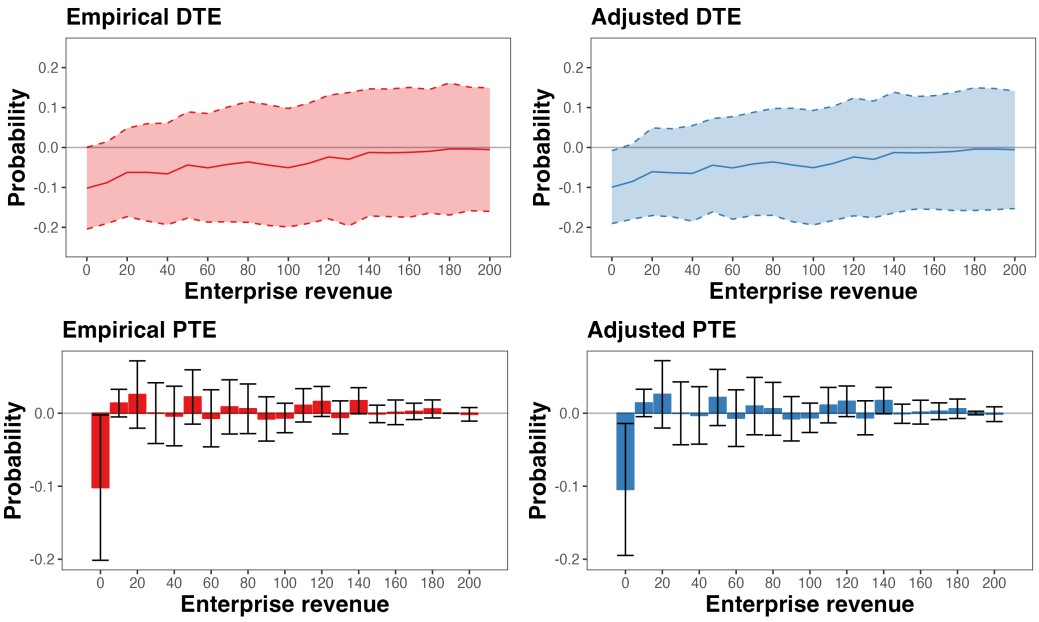

*Figure 3.* The Impacts of Microfinance: Distributional Treatment Effect (DTE) and Probability Treatment Effect (PTE) of joint liability lending on enterprise revenue (in thousand Mongolian Tugriks). The left panels depict empirical estimates, while the right panels present regression-adjusted estimates obtained using gradient boosting with 10-fold cross-fitting. Shaded regions and error bars represent 95% confidence intervals. Sample size: $n = 611$.

The experiment is one of many studies to evaluate the effectiveness of microcredit as a tool for alleviating global poverty. Since the launch of microcredit programs by the Grameen Bank in Bangladesh, which earned its founder Muhammad Yunus the Nobel Peace Prize in 2006, skepticism about their impact has grown in later years, fueled by the publication of findings on the subject. A primary goal of microcredit programs is to promote investment in and expansion of small-scale enterprises. However, Attanasio et al. (2015) found no significant average impact of these programs on enterprise revenue, profit, or other income sources.

In this paper, we revisit their analysis to estimate the distributional treatment effects of the lending program on enterprise revenue ($Y_i$), aiming to uncover potential heterogeneity beyond the average effects across the distribution. Our focus is on the comparison between joint-liability lending ($W_i = 2$) and the control group ($W_i = 1$), as this was the central concern of the original study. Notably, group lending was pioneered by the Grameen Bank in Bangladesh during the 1970s.

Figure 3 depicts the distributional and probability treatment effects of joint-liability lending on enterprise revenue. The outcome is measured in thousands of Mongolian Tugriks (MNT), with an exchange rate of 1 USD = 1150 MNT at the time of the study. We compute the DTE and PTE for $y \in \{0, 10, \ldots, 200\}$ accounting for the stratified design. For regression adjustment, we use gradient boosting with 10-fold cross-fitting, with pre-treatment covariates ($X_i$) including enterprise revenue prior to the experiment, household size, education level, age, etc. The full list of covariates can be found in Table 3 in the Appendix.

The top-left panel of Figure 3 presents the empirical DTE, while the top-right panel shows the regression-adjusted DTE. The shaded areas represent the 95% pointwise confidence band computed using multiplier bootstrap (Giné & Zinn, 1984; Belloni et al., 2017) with 1000 repetitions. Although the sample size in the experiment is modest ($n = 611$), the regression adjustment reduces the standard errors by 1% to 13% across revenue levels, with an average reduction of 7%. The bottom-left panel of Figure 3 presents the empirical PTE, and the bottom-right panel depicts the regression-adjusted PTE on enterprise revenue. For the PTE, the effectiveness of regression adjustment is limited by the small sample size, and it does not consistently reduce standard errors compared to the empirical estimator.

The analysis of DTE and PTE indicates that regression adjustment reduces the standard error by approximately 10% when estimating the probability of revenue being zero, leading to a statistically significant negative effect. Specifically, the probability of revenue being zero decreases by 10 percentage points (pp), with a standard error of 4.6 pp.

Overall, the evidence points to joint-liability lending mainly helping individuals move out of the "zero-revenue" trap, with little discernible effect elsewhere in the distribution. Economically, this pattern implies that the program acts more as a hedge against downside risk than as a catalyst for broad productivity gains, so welfare improvements are likely to flow from reduced business failure and smoother consumption rather than from large increases in aggregate output. Because the confidence bands remain wide beyond that lowest bin—even after flexible covariate adjustment—larger samples or richer, more predictive covariates will be needed to sharpen estimates across the rest of the revenue range.

## 7. Conclusion

We introduce a novel regression adjustment method designed to efficiently estimate distributional treatment effects under covariate-adaptive randomization. Our framework supports high-dimensional settings with many pre-treatment covariates and enables flexible modeling through the integration of off-the-shelf machine learning techniques for regression adjustment.

Despite its strengths, our method has certain limitations. First, it assumes experimental data with perfect compliance and no interference. While this setup is appropriate for some applications, it may limit applicability in contexts where these assumptions do not hold. Second, the effectiveness of our approach relies on the availability of pre-treatment covariates that are highly predictive of the outcome. Although we leverage flexible machine learning techniques to enhance prediction quality and achieve greater variance reduction compared to linear regression, the potential for variance reduction diminishes when covariates provide limited predictive power. Third, in scenarios with a large number of strata, improving precision using pre-treatment covariates becomes increasingly challenging, particularly when the sample size is modest. These limitations point to several promising directions for future research, such as designing methods to handle imperfect compliance and network effects, as well as enhancing estimation efficiency in settings with a large number of strata and locations. Additionally, extending recent advances in kernel mean embeddings to characterize outcome distributions (e.g., Park et al. (2021); Näf & Susmann (2024)) to the CAR framework represents a promising direction for future research.

## Acknowledgements

We extend our gratitude to the four anonymous reviewers and the program chairs for their insightful comments and discussions, which significantly enhanced the quality of this paper. Additionally, Oka acknowledges the financial support from JSPS KAKENHI Grant Number 24K04821.

## Impact Statement

This paper presents work whose goal is to advance the field of Machine Learning. There are many potential societal consequences of our work, none which we feel must be specifically highlighted here.

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

# Appendix

Appendix is organized as follows. Section A provides a table summarizing the notation. Section B elaborates on the Probability Treatment Effect. Section C presents all proofs. Finally, Section D offers additional details on the simulation study and real-data analysis.

## A. Summary of Notation

*Table 1.* Summary of Notation

| | |
|---|---|
| $X_i$ | pre-treatment covariates |
| $S_i$ | stratum indicator |
| $W_i$ | treatment variable |
| $Y_i$ | outcome variable |
| $Y_i(w)$ | potential outcome for treatment group $w$ |
| $p(s)$ | proportion of stratum $s$ |
| $\pi_w(s)$ | treatment assignment probability for treatment group $w$ in stratum $s$ |
| $n$ | sample size |
| $n_w(s)$ | number of observations in treatment group $w$ in stratum $s$ |
| $n(s)$ | number of observations in stratum $s$ |
| $\widehat{p}(s)$ | $n(s)/n$, proportion of stratum $s$ in the sample |
| $\widehat{\pi}_w(s)$ | $n_w(s)/n(s)$, estimated treatment assignment probability for treatment group $w$ in stratum $s$ |
| $F_{Y(w)}(y)$ | $\mathbb{E}[\mathbb{1}\{Y(w) \leq y\}]$, potential outcome distribution function |
| $[K]$ | $\{1, \ldots, K\}$ for a positive integer $K$ |
| $\|a\|$ | $\sqrt{a^\top a}$, Euclidean norm of a vector $a = (a_1, \ldots, a_p)^\top \in \mathbb{R}^p$ |
| $\|\cdot\|_{P,q}$ | $L_q(P)$ norm |
| $L^\infty(\mathcal{Y})$ | space of uniformly bounded functions mapping an arbitrary index set $\mathcal{Y}$ to the real line |
| $\rightsquigarrow$ | convergence in distribution or law |
| $\overset{d}{=}$ | equality in distribution |
| $X_n = O_p(a_n)$ | $\lim_{K \to \infty} \lim_{n \to \infty} P(|X_n| > K a_n) = 0$ for a sequence of positive constants $a_n$ |
| $X_n = o_p(a_n)$ | $\sup_{K > 0} \lim_{n \to \infty} P(|X_n| > K a_n) = 0$ for a sequence of positive constants $a_n$ |
| $x_n \lesssim y_n$ | for sequences $x_n$ and $y_n$ in $\mathbb{R}$, $x_n \leq A y_n$ for a constant $A$ that does not depend on $n$ |
| $\lfloor b \rfloor$ | $\max\{k \in \mathbb{Z} \mid k \leq b\}$, greatest integer less than or equal to $b$ |

## B. Probability Treatment Effect

Let $\rho_w(y_j, s, x) := P(y_{j-1} < Y(w) \le y_j | S = s, X = x) = \mathbb{E}[\mathbb{1}\{y_{j-1} < Y(w) \le y_j\} | S = s, X = x]$ and its estimator be denoted by $\widehat{\rho}_w(y_j, s, x)$. The empirical estimator for the bin probability for $y_j \in \mathcal{Y}$ is as follows:

$$
\begin{aligned}
\widehat{f}_{Y(w)}^{emp}(y_j) &:= \sum_{s=1}^{S} \widehat{p}(s) \Big\{ \frac{1}{n_w(s)} \sum_{i:W_i=w, S_i=s} \mathbb{1}\{y_{j-1} < Y_i \le y_j\} \Big\} \\
&= \frac{1}{n} \sum_{i=1}^{n} \frac{\mathbb{1}\{W_i = w\}}{\widehat{\pi}_w(S_i)} \cdot \mathbb{1}\{y_{j-1} < Y_i \le y_j\}.
\end{aligned}
\tag{B.1}
$$

Also, similarly, the regression-adjusted estimator for the bin probability for $y_j \in \mathcal{Y}_J$ is:

$$
\widehat{f}_{Y(w)}^{adj}(y_j) := \frac{1}{n} \sum_{i=1}^{n} \Big\{ \frac{\mathbb{1}\{W_i = w\}}{\widehat{\pi}_w(S_i)} \cdot \big(\mathbb{1}\{y_{j-1} < Y_i \le y_j\} - \widehat{\rho}_w(y_j, S_i, X_i)\big) + \widehat{\rho}_w(y_j, S_i, X_i) \Big\}.
\tag{B.2}
$$

Using these, we can define empirical and regression-adjusted PTE estimators as follows:

$$
\widehat{\Delta}_{w,w'}^{PTE,emp}(y_j) := \widehat{f}_{Y(w)}^{emp}(y_j) - \widehat{f}_{Y(w')}^{emp}(y_j),
\tag{B.3}
$$

$$
\widehat{\Delta}_{w,w'}^{PTE,adj}(y_j) := \widehat{f}_{Y(w)}^{adj}(y_j) - \widehat{f}_{Y(w')}^{adj}(y_j).
\tag{B.4}
$$

Then, the results in the paper also apply to the PTE, as the indicator functions $\mathbb{1}\{Y(w) \le y_j\}$ used in the analysis of DTE can be replaced with $\mathbb{1}\{y_{j-1} < Y(w) \le y_j\}$ for the analysis of the PTE.

## C. Proofs for Section 5

### C.1. Some definitions

We first introduce some definitions from empirical process theory that will be used in the proofs. See also van der Vaart & Wellner (1996) and Chernozhukov et al. (2014) for more details.

**Definition C.1** (Covering numbers). The *covering number* $N(\varepsilon, \mathcal{F}, \|\cdot\|)$ is the minimal number of balls $\{g : \|g - f\| < \varepsilon\}$ of radius $\varepsilon$ needed to cover the set $\mathcal{F}$. The centers of the balls need not belong to $\mathcal{F}$, but they should have finite norms.

**Definition C.2** (Envelope function). An *envelope function* of a class $\mathcal{F}$ is any function $x \mapsto F(x)$ such that $|f(x)| \le F(x)$ for every $x$ and $f$.

**Definition C.3** (VC-type class). We say $\mathcal{F}$ is of *VC-type* with coefficients $(\alpha, v)$ and envelope $F$ if the uniform covering numbers satisfy the following:

$$
\sup_{Q} N\left(\varepsilon \|F\|_{Q,2}, \mathcal{F}, L_2(Q)\right) \le \left(\frac{\alpha}{\varepsilon}\right)^v, \quad \forall \varepsilon \in (0, 1],
$$

where the supremum is taken over all finitely discrete probability measures.

### C.2. Asymptotic Distribution

***Proof of Theorem 5.2.*** Recall the notation $\mu_w(y, s) = \mathbb{E}[\mu_w(y, S_i, X_i) | S_i = s]$. Additionally, denote $\eta_{i,w}(y, s) := \mathbb{1}\{Y_i(w) \le y\} - \mu_w(y, s)$ and $D_w(s) := \sum_{i=1}^{n}(\mathbb{1}\{W_i = w\} - \pi_w(s)) \cdot \mathbb{1}\{S_i = s\}$. Note that we have $\widehat{\pi}_w(s) - \pi_w(s) = \frac{D_w(s)}{n(s)}$. We start with the linear expansion of $\widehat{F}_{Y(w)}^{adj}$ for treatment $w$ and decompose it into two terms $I_1(y)$ and $I_2(y)$ as

follows:

$$\sqrt{n}(\widehat{F}_{Y(w)}^{adj}(y) - F_{Y(w)}(y)) = \frac{1}{\sqrt{n}}\sum_{i=1}^{n}\left[\frac{\mathbb{1}\{W_i = w\}\cdot\left(\mathbb{1}\{Y_i \le y\} - \widehat{\mu}_w(y, S_i, X_i)\right)}{\widehat{\pi}_w(S_i)} + \widehat{\mu}_w(y, S_i, X_i)\right] - \sqrt{n}F_{Y(w)}(y)$$

$$= \underbrace{\frac{1}{\sqrt{n}}\sum_{i=1}^{n}\left[\frac{\mathbb{1}\{W_i = w\}}{\widehat{\pi}_w(S_i)}\cdot\mathbb{1}\{Y_i \le y\}\right] - \sqrt{n}F_{Y(w)}(y)}_{\equiv I_1(y)}$$

$$\underbrace{-\frac{1}{\sqrt{n}}\sum_{i=1}^{n}\left[\frac{(\mathbb{1}\{W_i = w\} - \widehat{\pi}_w(S_i))}{\widehat{\pi}_w(S_i)}\cdot\widehat{\mu}_w(y, S_i, X_i)\right]}_{\equiv I_2(y)}$$

Then, for the first term, we have

$$I_1(y) = \frac{1}{\sqrt{n}}\sum_{i=1}^{n}\sum_{s\in\mathcal{S}}\frac{\mathbb{1}\{W_i = w\}}{\pi_w(s)}\mathbb{1}\{S_i = s\}\mathbb{1}\{Y_i \le y\}$$

$$-\frac{1}{\sqrt{n}}\sum_{i=1}^{n}\sum_{s\in\mathcal{S}}\frac{\mathbb{1}\{W_i = w\}\mathbb{1}\{S_i = s\}(\widehat{\pi}_w(s) - \pi_w(s))}{\widehat{\pi}_w(s)\pi_w(s)}\mathbb{1}\{Y_i \le y\} - \sqrt{n}F_{Y(w)}(y)$$

$$= \frac{1}{\sqrt{n}}\sum_{i=1}^{n}\sum_{s\in\mathcal{S}}\frac{\mathbb{1}\{W_i = w\}}{\pi_w(s)}\mathbb{1}\{S_i = s\}\mathbb{1}\{Y_i \le y\}$$

$$-\sum_{i=1}^{n}\sum_{s\in\mathcal{S}}\frac{\mathbb{1}\{W_i = w\}\mathbb{1}\{S_i = s\}D_w(s)}{n(s)\sqrt{n}\widehat{\pi}_w(s)\pi_w(s)}\eta_{i,w}(y, s) - \sum_{s\in\mathcal{S}}\frac{D_w(s)\mu_w(y, s)}{n(s)\sqrt{n}\widehat{\pi}_w(s)\pi_w(s)}D_w(s) - \sum_{s\in\mathcal{S}}\frac{D_w(s)\mu_w(y, s)}{\sqrt{n}\widehat{\pi}_w(s)}$$

$$= \sum_{s\in\mathcal{S}}\frac{1}{\sqrt{n}}\sum_{i=1}^{n}\frac{\mathbb{1}\{W_i = w\}\mathbb{1}\{S_i = s\}}{\pi_w(s)}\eta_{i,w}(y, s) + \sum_{s\in\mathcal{S}}\frac{D_w(s)}{\sqrt{n}\pi_w(s)}\mu_w(y, s) + \sum_{i=1}^{n}\frac{\mu_w(y, S_i)}{\sqrt{n}}$$

$$-\sum_{i=1}^{n}\sum_{s\in\mathcal{S}}\frac{\mathbb{1}\{W_i = w\}\mathbb{1}\{S_i = s\}D_w(s)}{n(s)\sqrt{n}\widehat{\pi}_w(s)\pi_w(s)}\eta_{i,w}(y, s) - \sum_{s\in\mathcal{S}}\frac{D_w(s)\mu_w(y, s)}{n(s)\sqrt{n}\widehat{\pi}_w(s)\pi_w(s)}D_w(s) - \sum_{s\in\mathcal{S}}\frac{D_w(s)\mu_w(y, s)}{\sqrt{n}\widehat{\pi}_w(s)}$$

$$= \sum_{s\in\mathcal{S}}\frac{1}{\sqrt{n}}\sum_{i=1}^{n}\frac{\mathbb{1}\{W_i = w\}\mathbb{1}\{S_i = s\}}{\pi_w(s)}\eta_{i,w}(y, s) + \sum_{i=1}^{n}\frac{\mu_w(y, S_i)}{\sqrt{n}} + I_{1,1}(y),$$

where

$$I_{1,1}(y) = -\sum_{i=1}^{n}\sum_{s\in\mathcal{S}}\frac{\mathbb{1}\{W_i = w\}1\{S_i = s\}D_w(s)}{n(s)\sqrt{n}\widehat{\pi}_w(s)\pi_w(s)}\eta_{i,w}(y, s) - \sum_{s\in\mathcal{S}}\frac{D_w(s)\mu_w(y, s)}{n(s)\sqrt{n}\widehat{\pi}_w(s)\pi_w(s)}D_w(s)$$

$$+ \sum_{s\in\mathcal{S}}\frac{D_w(s)\mu_w(y, s)}{\sqrt{n}}\left(\frac{1}{\pi_w(s)} - \frac{1}{\widehat{\pi}_w(s)}\right)$$

$$= -\sum_{i=1}^{n}\sum_{s\in\mathcal{S}}\frac{\mathbb{1}\{W_i = w\}\mathbb{1}\{S_i = s\}D_w(s)}{n(s)\sqrt{n}\widehat{\pi}_w(s)\pi_w(s)}\eta_{i,w}(y, s).$$

Note that the class $\{\mathbb{1}\{Y_i(w) \le y\} - \mu_w(y, S_i) : y \in \mathcal{Y}\}$ is of the VC-type with fixed coefficients $(\alpha, v)$ and bounded envelope, and $\mathbb{E}[\mathbb{1}\{Y_i(w) \le y\} - \mu_w(y, S_i)|S_i = s] = 0$. Therefore,

$$\sup_{y\in\mathcal{Y},s\in\mathcal{S}}\left|\frac{1}{\sqrt{n}}\sum_{i=1}^{n}\mathbb{1}\{W_i = w\}\mathbb{1}\{S_i = s\}\eta_{i,w}(y, s)\right| = O_p(1).$$

By Assumption 3.1, for all $w \in \mathcal{W}$, we have $\max_{s\in\mathcal{S}}|D_w(s)/n(s)| = o_p(1)$, $\max_{s\in\mathcal{S}}|\widehat{\pi}_w(s) - \pi_w(s)| = o_p(1)$, and $\min_{s\in\mathcal{S}}\pi_w(s) > c > 0$, which imply $\sup_{y\in\mathcal{Y}}|I_{1,1}(y)| = o_p(1)$.

Next, we analyze the second term $I_2(y)$.

$$I_2(y) = \frac{1}{\sqrt{n}} \sum_{s \in \mathcal{S}} \sum_{i=1}^{n} \frac{\mathbb{1}\{W_i = w\}}{\hat{\pi}_w(s)} \mu_w(y, s, X_i) \mathbb{1}\{S_i = s\} - \frac{1}{\sqrt{n}} \sum_{i=1}^{n} \mu_w(y, S_i, X_i)$$

$$+ \frac{1}{\sqrt{n}} \sum_{s \in \mathcal{S}} \frac{1}{\hat{\pi}_w(s)} \sum_{i=1}^{n} (\mathbb{1}\{W_i = w\} - \hat{\pi}_w(s)) (\hat{\mu}_w(y, s, X_i) - \mu_w(y, s, X_i)) \mathbb{1}\{S_i = s\}$$

$$= \frac{1}{\sqrt{n}} \sum_{s \in \mathcal{S}} \sum_{i=1}^{n} \frac{\mathbb{1}\{W_i = w\}}{\hat{\pi}_w(s)} (\mu_w(y, s, X_i) - \mu_w(y, s)) \mathbb{1}\{S_i = s\} - \frac{1}{\sqrt{n}} \sum_{i=1}^{n} (\mu_w(y, S_i, X_i) - \mu_w(y, S_i))$$

$$+ \frac{1}{\sqrt{n}} \sum_{s \in \mathcal{S}} \frac{1}{\hat{\pi}_w(s)} \sum_{i=1}^{n} (\mathbb{1}\{W_i = w\} - \hat{\pi}_w(s)) (\hat{\mu}_w(y, s, X_i) - \mu_w(y, s, X_i)) \mathbb{1}\{S_i = s\}$$

$$= \frac{1}{\sqrt{n}} \sum_{s \in \mathcal{S}} \sum_{i=1}^{n} \frac{\mathbb{1}\{W_i = w\}}{\pi_w(s)} (\mu_w(y, s, X_i) - \mu_w(y, s)) \mathbb{1}\{S_i = s\} - \frac{1}{\sqrt{n}} \sum_{i=1}^{n} (\mu_w(y, S_i, X_i) - \mu_w(y, S_i))$$

$$+ \underbrace{\frac{1}{\sqrt{n}} \sum_{s \in \mathcal{S}} \left( \frac{\pi_w(s) - \hat{\pi}_w(s)}{\hat{\pi}_w(s) \pi_w(s)} \right) \left( \sum_{i=1}^{n} \mathbb{1}\{W_i = w\}(\mu_w(y, s, X_i) - \mu_w(y, s)) \mathbb{1}\{S_i = s\} \right)}_{\equiv I_{2,1}(y)}$$

$$+ \underbrace{\frac{1}{\sqrt{n}} \sum_{s \in \mathcal{S}} \frac{1}{\hat{\pi}_w(s)} \sum_{i=1}^{n} (\mathbb{1}\{W_i = w\} - \hat{\pi}_w(s)) (\hat{\mu}_w(y, s, X_i) - \mu_w(y, s, X_i)) \mathbb{1}\{S_i = s\}}_{\equiv I_{2,2}(y)}$$

where the second equality holds because

$$\sum_{s \in \mathcal{S}} \sum_{i=1}^{n} \frac{\mathbb{1}\{W_i = w\}}{\hat{\pi}_w(s)} \mu_w(y, s) \mathbb{1}\{S_i = s\} = \sum_{s \in \mathcal{S}} n(s) \mu_w(y, s) = \sum_{i=1}^{n} \mu_w(y, S_i).$$

For the first term $I_{2,1}(y)$, we have

$$\sup_{y \in \mathcal{Y}} \left| \frac{1}{\sqrt{n}} \sum_{s \in \mathcal{S}} \left( \frac{\pi_w(s) - \hat{\pi}_w(s)}{\hat{\pi}_w(s) \pi_w(s)} \right) \left( \sum_{i=1}^{n} \mathbb{1}\{W_i = w\}(\mu_w(y, s, X_i) - \mu_w(y, s)) \mathbb{1}\{S_i = s\} \right) \right|$$

$$\leq \sum_{s \in \mathcal{S}} \left| \frac{D_w(s)}{n_w(s) \pi_w(s)} \right| \sup_{y \in \mathcal{Y}, s \in \mathcal{S}} \left| \frac{1}{\sqrt{n}} \sum_{i=1}^{n} \mathbb{1}\{W_i = w\} \mathbb{1}\{S_i = s\}(\mu_w(y, s, X_i) - \mu_w(y, s)) \right|.$$

Assumption 5.1 implies that the class $\{\mu_w(y, s, X_i) - \mu_w(y, s) : y \in \mathcal{Y}\}$ is of the VC-type with fixed coefficients $(\alpha, v)$ and an envelope $F_i$ such that $\mathbb{E}(|F_i|^d | S_i = s) < \infty$ for $d > 2$. Therefore,

$$\sup_{y \in \mathcal{Y}, s \in \mathcal{S}} \left| \frac{1}{\sqrt{n}} \sum_{i=1}^{n} \mathbb{1}\{W_i = w\} \mathbb{1}\{S_i = s\}(\mu_w(y, s, X_i) - \mu_w(y, s)) \right| = O_p(1).$$

It is also assumed that $D_w(s)/n(s) = o_p(1)$ and $n(s)/n_w(s) \xrightarrow{p} 1/\pi_w(s) < \infty$. Therefore, we have

$$\sup_{y \in \mathcal{Y}} |I_{2,1}(y)| = o_p(1).$$

As for the second term $I_{2,2}(y)$, using the notation $\delta_w(y, s, X_i) = \hat{\mu}_w(y, s, X_i) - \mu_w(y, s, X_i)$, by Assumption 5.1(i), we

have

$$\sup_{y\in\mathcal{Y}}\left|\frac{1}{\sqrt{n}}\sum_{s\in\mathcal{S}}\frac{1}{\hat{\pi}_w(s)}\sum_{i=1}^{n}(\mathbb{1}\{W_i=w\}-\hat{\pi}_w(s))\delta_w(y,s,X_i)\mathbb{1}\{S_i=s\}\right|$$

$$\leq\frac{1}{\sqrt{n}}\sum_{s\in\mathcal{S}}n(s)\sup_{y\in\mathcal{Y}}\left|\frac{\sum_{i\in I_w(s)}\delta_w(y,s,X_i)}{n_w(s)}-\frac{\sum_{i\in I_{w'}(s)}\delta_w(y,s,X_i)}{n_{w'}(s)}\right|=o_p(1).$$

Therefore, we have

$$\sup_{y\in\mathcal{Y}}|I_{2,1}(y)+I_{2,2}(y)|=o_p(1).$$

Combining the two terms, we have

$$\sqrt{n}(\widehat{F}_{Y(w)}^{adj}(y)-F_{Y(w)}(y))=\sum_{s\in\mathcal{S}}\frac{1}{\sqrt{n}}\sum_{i=1}^{n}\mathbb{1}\{W_i=w\}\mathbb{1}\{S_i=s\}\left[\frac{\eta_{i,w}(y,s)}{\pi_w(s)}+\left(1-\frac{1}{\pi_w(s)}\right)(\mu_w(y,s,X_i)-\mu_w(y,s))\right]$$

$$+\sum_{s\in\mathcal{S}}\frac{1}{\sqrt{n}}\sum_{i=1}^{n}(1-\mathbb{1}\{W_i=w\})\mathbb{1}\{S_i=s\}(\mu_w(y,s,X_i)-\mu_w(y,s))$$

$$+\sum_{i=1}^{n}\frac{\mu_w(y,S_i)}{\sqrt{n}}+R_{1,1}(y),$$

where $\sup_{y\in\mathcal{Y}}|R_{1,1}(y)|=o_p(1)$.

Define

$$\phi_w(y,S_i,Y_i(w),X_i):=\frac{\eta_{i,w}(y,s)}{\pi_w(s)}+\left(1-\frac{1}{\pi_w(s)}\right)(\mu_w(y,s,X_i)-\mu_w(y,s))-(\mu_{w'}(y,s,X_i)-\mu_{w'}(y,s)),\quad\text{(C.5)}$$

$$\zeta_i(y):=\mu_w(y,S_i)-\mu_{w'}(y,S_i).$$

Taking the difference between treatments $w$ and $w'$, we obtain

$$\sqrt{n}\big(\widehat{\Delta}_{w,w'}^{DTE,adj}(y)-\Delta_{w,w'}^{DTE}(y)\big)=\sum_{s\in\mathcal{S}}\frac{1}{\sqrt{n}}\sum_{i=1}^{n}\mathbb{1}\{W_i=w\}\mathbb{1}\{S_i=s\}\phi_w(y,S_i,Y_i(w),X_i)$$

$$-\sum_{s\in\mathcal{S}}\frac{1}{\sqrt{n}}\sum_{i=1}^{n}\mathbb{1}\{W_i=w'\}\mathbb{1}\{S_i=s\}\phi_{w'}(y,S_i,Y_i(w'),X_i)$$

$$+\frac{1}{\sqrt{n}}\sum_{i=1}^{n}\zeta_i(y)+R(y),$$

where $\sup_{y\in\mathcal{Y}}|R(y)|=o_p(1)$. Denote

$$\varphi_{n,1}(y):=\sum_{s\in\mathcal{S}}\frac{1}{\sqrt{n}}\sum_{i=1}^{n}\mathbb{1}\{W_i=w,S_i=s\}\phi_w(y,s,Y_i(w),X_i)-\sum_{s\in\mathcal{S}}\frac{1}{\sqrt{n}}\sum_{i=1}^{n}\mathbb{1}\{W_i=w',S_i=s\}\phi_{w'}(y,s,Y_i(w'),X_i),$$

and

$$\varphi_{n,2}(y):=\frac{1}{\sqrt{n}}\sum_{i=1}^{n}\zeta_i(y).$$

Then, uniformly over $y\in\mathcal{Y}$, we first show that

$$(\varphi_{n,1}(y),\varphi_{n,2}(y))\rightsquigarrow(\mathcal{G}_1(y),\mathcal{G}_2(y)),$$

where $(\mathcal{G}_1(y), \mathcal{G}_2(y))$ are two independent Gaussian processes with covariance kernels $\Omega_1(y, y')$ and $\Omega_2(y, y')$, respectively, such that

$$\Omega_1(y, y') = \mathbb{E}[\pi_w(S_i)\phi_w(y, S_i, Y_i(w), X_i)\phi_w(y', S_i, Y_i(w), X_i)]$$
$$+ \mathbb{E}[\pi_{w'}(S_i)\phi_{w'}(y, S_i, Y_i(w'), X_i)\phi_{w'}(y', S_i, Y_i(w'), X_i)],$$

and

$$\Omega_2(y, y') = \mathbb{E}[\zeta_i(y)\zeta_i(y')].$$

The following argument follows the argument provided in the proofs of Bugni et al. (2018, Lemma B.2) and Bugni et al. (2019, Lemma C.1). We first argue that, uniformly over $y \in \mathcal{Y}$,

$$(\varphi_{n,1}(y), \varphi_{n,2}(y)) \rightsquigarrow (\varphi^\star_{n,1}(y), \varphi_{n,2}(y)),$$

where $\varphi^\star_{n,1}(y) \perp\!\!\!\perp \varphi_{n,2}(y)$ and $\varphi^\star_{n,1}(y) \rightsquigarrow \mathcal{G}_1(y)$ uniformly over $y \in \mathcal{Y}$.

Under Assumption 3.1, the distribution of $\varphi_{n,1}(y)$ is the same as the distribution of the same quantity with units ordered by strata $s \in \mathcal{S}$ and then ordered by treatment assignment $w \in \mathcal{W}$ within strata. In order to exploit this observation, we introduce the following notations. Define $N(s) := \sum_{i=1}^n \mathbb{1}\{S_i < s\}$, $N_w(s) := \sum_{i=1}^n \mathbb{1}\{W_i < w, S_i = s\}$, $F(s) := \mathbb{P}(S_i < s)$, and $F_w(s) := \mathbb{P}(W_i < w, S_i = s)$ for all $(w, s) \in \mathcal{W} \times \mathcal{S}$. Furthermore, let $\{(X_i^s, Y_i^s(1), \ldots, Y_i^s(|\mathcal{W}|)) : 1 \leq i \leq n\}$ be a sequence of i.i.d. random variables with marginal distributions equal to the distribution of $(X_i, Y_i(1), \ldots, Y_i(|\mathcal{W}|))|S_i = s$ and

$$\varphi_{n,1}(y)|\{(W_i, S_i)_{i \in [n]}\} \overset{d}{=} \widetilde{\varphi}_{n,1}(y)|\{(W_i, S_i)_{i \in [n]}\}$$

where

$$\widetilde{\varphi}_{n,1}(y) := \sum_{s \in \mathcal{S}} \frac{1}{\sqrt{n}} \sum_{i=N(s)+N_w(s)+1}^{N(s)+N_{w+1}(s)} \phi_w(y, s, Y_i^s(w), X_i^s) - \sum_{s \in \mathcal{S}} \frac{1}{\sqrt{n}} \sum_{i=N(s)+N_{w'}(s)+1}^{N(s)+N_{w'+1}(s)} \phi_{w'}(y, s, Y_i^s(w'), X_i^s).$$

As $\varphi_{n,2}(y)$ is only a function of $\{S_i\}_{i \in [n]}$, we have

$$(\varphi_{n,1}(y), \varphi_{n,2}(y)) \overset{d}{=} (\widetilde{\varphi}_{n,1}(y), \varphi_{n,2}(y)).$$

Next, define

$$\varphi^\star_{n,1}(y) := \sum_{s \in \mathcal{S}} \frac{1}{\sqrt{n}} \sum_{i=\lfloor nF(s)+F_w(s)\rfloor+1}^{\lfloor n(F(s)+F_{w+1}(s))\rfloor} \phi_w(y, s, Y_i^s(w), X_i^s) - \sum_{s \in \mathcal{S}} \frac{1}{\sqrt{n}} \sum_{i=\lfloor n(F(s)+F_{w'}(s))\rfloor+1}^{\lfloor n(F(s)+F_{w'+1}(s))\rfloor} \phi_{w'}(y, s, Y_i^s(w'), X_i^s).$$

Note $\varphi^\star_{n,1}(y)$ is a function of $(Y_i^s(1), \ldots, Y_i^s(|\mathcal{W}|), X_i^s)_{i \in [n], s \in \mathcal{S}}$, which is independent of $\{W_i, S_i\}_{i \in [n]}$ by construction. Therefore,

$$\varphi^\star_{n,1}(y) \perp\!\!\!\perp \varphi_{n,2}(y).$$

Note that

$$\frac{N(s)}{n} \overset{p}{\longrightarrow} F(s), \quad \frac{N_w(s)}{n} \overset{p}{\longrightarrow} F_w(s), \quad \text{and} \quad \frac{n(s)}{n} \overset{p}{\longrightarrow} p(s).$$

We shall show that

$$\sup_{y \in \mathcal{Y}} |\widetilde{\varphi}_{n,1}(y) - \varphi^\star_{n,1}(y)| = o_p(1) \quad \text{and} \quad \varphi^\star_{n,1}(y) \rightsquigarrow \mathcal{G}_1(y).$$

Since $\mathcal{S}$ and $\mathcal{W}$ have finite cardinality, we fix $(s, w) \in \mathcal{S} \times \mathcal{W}$ in the remainder of the proof. We define

$$\Gamma_n(t, \phi_w) := \frac{1}{\sqrt{n}} \sum_{i=1}^n \mathbb{1}\{i \leq \lfloor nt \rfloor\} \cdot \phi_w(y, s, Y_i^s(w), X_i^s),$$

for $t \in (0, 1]$. The function $\phi_w(y, s, Y_i^s(w), X_i^s)$ defined in (C.5) can be decomposed as a weighted sum of bounded random functions indexed by $y \in \mathcal{Y}$ with bounded weight functions. More precisely, the class $\mathcal{F} := \{\phi_w(y, s, Y_i^s(w), X_i^s) : y \in \mathcal{Y}\}$ consists of functions from the following function classes: $\mathcal{F}_1 := \{y \mapsto \eta_{i,w}(y, s)\}$, $\mathcal{F}_{2,w} := \{y \mapsto \mu_w(y, s, X_i)\}$ and $\mathcal{F}_{3,w} := \{y \mapsto \mu_w(y, s)\}$. We can show that the class $\mathcal{F}_1$ is Donsker, for instance, by using the bounded, monotone property as established in Theorem 2.7.5 of van der Vaart & Wellner (1996). Also, under Assumption 5.1(ii), Theorem 2.5.2 of van der Vaart & Wellner (1996) yields that $\mathcal{F}_{2,w}$ and $\mathcal{F}_{3,w}$ are Donsker. Since all the random weights are uniformly bounded, Corollary 2.10.13 of van der Vaart & Wellner (1996) shows that $\mathcal{F}$ is Donsker. Also, the class $\{t \mapsto \mathbb{1}\{i \leq \lfloor nt \rfloor\}\}$ is VC class and hence Donsker. Since Theorem 2.10.6 of van der Vaart & Wellner (1996) shows that products of uniformly bounded Donsker classes are Donsker, we conclude that the indexed process $\{\Gamma_n(t, \phi_w) : t \in (0, 1], \phi_w \in \mathcal{F}\}$ is Donsker. Hence, the result follows.

Next, for a given $y$, by the triangular array central limit theorem,

$$\varphi_{n,1}^\star(y) \rightsquigarrow N(0, \Omega_1(y, y)),$$

where

$$
\begin{aligned}
\Omega_1(y, y) &= \lim_{n \to \infty} \sum_{s \in \mathcal{S}} \frac{(\lfloor n(F(s) + F_{w+1}(s)) \rfloor - \lfloor n(F(s) + F_w(s)) \rfloor)}{n} \mathbb{E}[\phi_w^2(y, s, Y_i^s(w), X_i^s)] \\
&+ \lim_{n \to \infty} \sum_{s \in \mathcal{S}} \frac{(\lfloor n(F(s) + F_{w'+1}(s)) \rfloor - \lfloor n(F(s) + F_{w'}(s)) \rfloor)}{n} \mathbb{E}[\phi_{w'}^2(y, s, Y_i^s(w'), X_i^s)] \\
&= \sum_{s \in \mathcal{S}} p(s) \mathbb{E}[\pi_w(s) \phi_w^2(y, S_i, Y_i(w), X_i) + \pi_{w'}(s) \phi_{w'}^2(y, S_i, Y_i(w'), X_i) | S_i = s] \\
&= \mathbb{E}[\pi_w(S_i) \phi_w^2(y, S_i, Y_i(w), X_i)] + \mathbb{E}[\pi_{w'}(S_i) \phi_{w'}^2(y, S_i, Y_i(w'), X_i)].
\end{aligned}
$$

Using the Cramér–Wold device to verify finite-dimensional convergence, we find that the covariance kernel can be written as

$$
\begin{aligned}
\Omega_1(y, y') =& \mathbb{E}[\pi_w(S_i) \phi_w(y, S_i, Y_i(w), X_i) \phi_w(y', S_i, Y_i(w), X_i)] \\
&+ \mathbb{E}[\pi_{w'}(S_i) \phi_{w'}(y, S_i, Y_i(w'), X_i) \phi_{w'}(y', S_i, Y_i(w'), X_i)].
\end{aligned}
$$

Finally, since $\{\mu_w(y, S_i) : y \in \mathcal{Y}\}$ is of the VC-type with fixed coefficients $(\alpha, v)$ and a constant envelope function, $\{\mu_w(y, S_i) - \mu_{w'}(y, S_i) : y \in \mathcal{Y}\}$ is a Donsker class and we have

$$\varphi_{n,2}(y) \rightsquigarrow \mathcal{G}_2(y),$$

where $\mathcal{G}_2(y)$ is a Gaussian process with covariance kernel

$$\Omega_2(y, y') = \mathbb{E}[(\mu_w(y, S_i) - \mu_{w'}(y, S_i))(\mu_w(y', S_i) - \mu_{w'}(y', S_i))] \equiv \mathbb{E}[\zeta_i(y)\zeta_i(y')].$$

Therefore, combining the results, we have, uniformly over $y \in \mathcal{Y}$,

$$\sqrt{n}(\widehat{\Delta}_{w,w'}^{DTE,adj}(y) - \Delta_{w,w'}^{DTE}(y)) \rightsquigarrow \mathcal{G}(y),$$

where $\mathcal{G}(y)$ is a Gaussian process with covariance kernel

$$
\begin{aligned}
\Omega(y, y') =& \mathbb{E}[\pi_w(S_i) \phi_w(y, S_i, Y_i(w), X_i) \phi_w(y', S_i, Y_i(w), X_i)] \\
&+ \mathbb{E}[\pi_{w'}(S_i) \phi_{w'}(y, S_i, Y_i(w'), X_i) \phi_{w'}(y', S_i, Y_i(w'), X_i)] \\
&+ \mathbb{E}[\zeta_i(y)\zeta_i(y')].
\end{aligned}
$$

This concludes the proof. $\qquad\square$

## C.3. Semiparametric Efficiency Bound

***Proof of Theorem 5.3.*** Part (a). We follow the approach used in Hahn (1998) and calculate the semiparametric efficiency bound of the DTE: $\Delta_{w,w'}^{DTE}(u)$ for a given $u \in \mathcal{Y}$. Throughout the proof of part (a), we use $u$ to indicate this evaluation location to keep the notation clear. First, we characterize the tangent space. To that end, the joint density of the observed variables $(Y, W, X, S)$ can be written as:

$$f(y, w, x, s) = f(y|w, x, s)f(w|x, s)f(x|s)f(s) = \prod_{w=1}^{K} \{f_w(y|x, s)\pi_w(s)\}^{\mathbb{1}\{W=w\}} f(x|s)f(s),$$

where $f_w(y|x, s) := P(Y = y|W = w, X = x, S = s)$ and $\pi_w(s) := P(W = w|X = x, S = s)$ for all $x \in \mathcal{X}$.

Consider a regular parametric submodel indexed by $\theta$:

$$f(y, w, x, s; \theta) := \prod_{w=1}^{K} \{f_w(y|x, s; \theta)\pi_w(s; \theta)\}^{\mathbb{1}\{W=w\}} f(x, s; \theta)f(s; \theta),$$

which equals $f(y, w, x, s)$ when $\theta = \theta_0$.

The corresponding score of $f(y, w, x; \theta)$ is given by

$$s(y, w, x, s; \theta) := \frac{\partial \ln f(y, w, x, s; \theta)}{\partial \theta}$$

$$= \sum_{w=1}^{K} \left( \mathbb{1}\{W = w\}\dot{f}_w(y|x, s; \theta) + \mathbb{1}\{W = w\}\dot{\pi}_w(s; \theta) \right) + \dot{f}(x, s; \theta) + \dot{f}(s; \theta),$$

where $\dot{f}$ denotes a derivative of the log, i.e, $\dot{f}(x; \theta) = \frac{\partial \ln f(x;\theta)}{\partial \theta}$.

At the true value, the expectation of the score equals zero. The tangent space of the model is the set of functions that are mean zero and satisfy the additive structure of the score:

$$\mathcal{T} = \left\{ \sum_{w=1}^{K} \left( \mathbb{1}\{W = w\}a_w(y|x, s) + \mathbb{1}\{W = w\}a_\pi(s) \right) + a_x(x, s) + a_s(s) \right\}, \tag{C.6}$$

where $a_w(y|x, s)$, $a_\pi(s)$, $a_x(x, s)$ and $a_s(s)$ are mean-zero functions.

The semiparametric variance bound of $\Delta_{w,w'}^{DTE}(u)$ is the variance of the projection on $\mathcal{T}$ of a function $\psi_u(Y, W, X, S)$ (with mean zero and finite second order moment) that satisfies for all regular parametric submodels

$$\left.\frac{\partial \Delta_{w,w'}^{DTE}(u; F_\theta)}{\partial \theta}\right|_{\theta=\theta_0} = \mathbb{E}[\psi_u(Y, W, X, S) \cdot s(Y, W, X, S)]\Big|_{\theta=\theta_0} \tag{C.7}$$

If $\psi_u$ itself already lies in the tangent space, the variance bound is given by $\mathbb{E}[\psi_u^2]$.

Now, the DTE is

$$\Delta_{w,w'}^{DTE}(u; F_\theta) = \iiint \mathbb{1}\{y \le u\}f_w(y|x, s; \theta)f(x|s; \theta)f(s; \theta)dydxds$$

$$- \iiint \mathbb{1}\{y \le u\}f_{w'}(y|x, s; \theta)f(x|s; \theta)f(s; \theta)dydxds.$$

We thus have

$$
\begin{aligned}
\frac{\partial \Delta_{w,w'}^{DTE}(u; F_\theta)}{\partial \theta}\Big|_{\theta=\theta_0} =& \iiint \mathbb{1}\{y \le u\} \dot{f}_w(y|x,s;\theta) f_w(y|x,s) f(x|s) f(s) dy dx ds \\
&+ \iint \mu_w(u,s,x) \dot{f}(x|s;\theta) f(x|s) f(s) dx ds \\
&+ \int \mu_w(u,s,x) f(x|s) \dot{f}(s;\theta) f(s) ds \\
&- \iiint \mathbb{1}\{y \le u\} \dot{f}_{w'}(y|x,s;\theta) f_{w'}(y|x,s) f(x|s) f(s) dy dx ds \\
&- \iint \mu_{w'}(u,s,x)) \dot{f}(x|s;\theta) f(x|s) f(s) dx ds \\
&- \int \mu_{w'}(u,s,x) f(x|s) \dot{f}(s;\theta) f(s) ds.
\end{aligned}
$$

Letting $\mu_{w,w'}(u,S,X) := \mu_w(u,S,X) - \mu_{w'}(u,S,X)$, we choose $\psi_u(Y,W,X,S)$ as

$$
\begin{aligned}
\psi_u(Y,W,X,S) =& \frac{\mathbb{1}\{W=w\}}{\pi_w(S)}(\mathbb{1}\{Y \le u\} - \mu_w(u,S,X)) - \frac{\mathbb{1}\{W=w'\}}{\pi_{w'}(S)}(\mathbb{1}\{Y \le u\} - \mu_{w'}(u,S,X)) \\
&+ \mu_{w,w'}(u,S,X) - \Delta_{w,w'}^{DTE}(u).
\end{aligned}
$$

Notice that $\psi_u$ satisfies equation (C.7) and that $\psi_u$ lies in the tangent space $\mathcal{T}$ given in equation (C.6). Since $\psi_u$ lies in the tangent space, the variance bound is given by the expected square of $\psi_u$:

$$
\begin{aligned}
\Omega(u) :=& \mathbb{E}\big[\psi_u(Y,W,X,S)^2\big] \\
=& \mathbb{E}\Big[\Big(\frac{\mathbb{1}\{W=w\}}{\pi_w(S)}(\mathbb{1}\{Y \le u\} - \mu_w(u,S,X)) - \frac{\mathbb{1}\{W=w'\}}{\pi_{w'}(S)}(\mathbb{1}\{Y \le u\} - \mu_{w'}(u,S,X)) \\
&+ \mu_{w,w'}(u,S,X) - \Delta_{w,w'}^{DTE}(u)\Big)^2\Big] \\
=& \mathbb{E}[\pi_w(S)\phi_w^2(u,S,Y(w),X)] + \mathbb{E}[\pi_{w'}(S)\phi_{w'}^2(u,S,Y(w'),X)] + \mathbb{E}[\zeta_i^2(u)].
\end{aligned}
$$

This concludes the proof of part (a).

Next, for part (b), under Assumption 5.1, the regression-adjusted estimator defined in Algorithm 1 satisfies the following asymptotic distribution for any given $y \in \mathcal{Y}$:

$$
\sqrt{n}\big(\widehat{\Delta}_{w,w'}^{DTE,adj}(y) - \Delta_{w,w'}^{DTE}(y)\big) \rightsquigarrow \mathcal{N}(0, \Omega(y)),
$$

where $\Omega(y)$ is the semiparametric efficiency bound derived in part (a). This completes the proof of part (b). □

***Proof of Corollary 5.4.*** The result follows directly from the fact that, for a given $y \in \mathcal{Y}$, $\widehat{\Delta}_{w,w'}^{DTE,emp}(y)$ is a regular, consistent and asymptotically normal estimator for the DTE. Moreover, the variance of $\tilde{\Delta}_{w,w'}^{DTE,adj}(y)$ coincides with the semiparametric efficiency bound $\Omega(y)$ established in Theorem 5.3. Consequently, we obtain

$$
Var\big(\tilde{\Delta}_{w,w'}^{DTE,adj}(y)\big) \le Var\big(\widehat{\Delta}_{w,w'}^{DTE,emp}(y)\big),
$$

which concludes the proof. □

# D. Additional experimental details

All experiments were carried out on a MacBook Pro equipped with an Apple M3 Pro chip and 36GB memory. The replication code is publicly available at https://github.com/CyberAgentAILab/dte_car, and the method can be implemented using the Python library `dte-adj` (https://pypi.org/project/dte-adj/).

## D.1. Simulation Study

Figure 4 illustrates the percentage reduction in RMSE of regression-adjusted estimators relative to the empirical estimator for different sample sizes: $n \in \{1,000, 5,000, 10,000\}$. Across all outcome levels, ML adjustment consistently outperforms linear regression, achieving reductions in RMSE of up to 50%. The RMSE reduction increases notably as the sample size grows from 1,000 to 5,000, but the improvement is less pronounced when the sample size increases from 5,000 to 10,000.

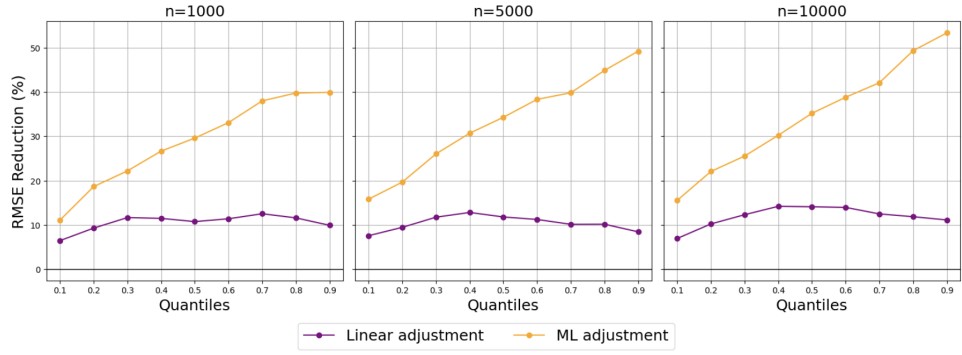

*Figure 4.* RMSE reduction (%) of regression-adjusted estimators vs. empirical estimator across quantiles for continuous outcomes, with $n \in \{1,000, 5,000, 10,000\}$. Linear adjustment uses linear regression; ML adjusment uses gradient boosting. Both use 2-fold cross-fitting. Number of simulations is 1,000.

### D.1.1. Additional Simulation Results

We consider a discrete outcome $Y_i$ that follows a Poisson distribution with the conditional expected value given by:

$$E[Y_i|X_i, W_i, Z_i] = 0.2 \left| b(X_i) + c(X_i)W_i + \gamma Z_i \right|,$$

where $b(X_i)$ and $c(X_i)$ are given in Section 6.1. All other variables and treatment assignment mechanism are consistent with those described in Section 6.1. Figure 5 presents the RMSE of the DTE estimators, along with the average length and coverage probabilities of their 95% confidence intervals, for the empirical estimator and the regression-adjusted estimators using both linear and machine learning (ML) adjustments, with a sample size of $n = 1,000$. Figure 6 shows the percentage reduction in RMSE of the regression-adjusted estimators relative to the empirical estimator across varying sample sizes: $n \in \{1,000, 5,000, 10,000\}$. Similar to the results observed in the continuous outcome simulation, ML adjustment consistently outperforms linear regression. The benefit of regression adjustment increases with sample size: while neither method achieves RMSE reduction for large values of $Y$ when $n = 1,000$, both show significant RMSE reductions across all outcome levels when $n = 10,000$.

We compare our method with the quantile treatment effect (QTE) estimator by Jiang et al. (2023) in the simulation setting described in Section 6.1, using linear regression. Additionally, we examine a scenario with discrete covariates, which leads to a discrete outcome variable. In this setting, we retain the same data-generating process but sample each covariate $X_i$ independently from a Uniform$(-5, 5)$ distribution and round the values to the nearest integer. Table 2 presents the results, showing that our proposed method with linear adjustment achieves a reduction in RMSE ranging from 6.4% to 12.5% for the continuous outcome and from 0.8% to 6.1% for the discrete outcome. In contrast, the method by Jiang et al. (2023) achieves up to a 6.5% reduction in RMSE for the continuous outcome but does not yield improvements for the discrete outcome. Furthermore, the DTE estimator demonstrates substantially better computational efficiency. It requires only 0.008 seconds for the continuous outcome and 0.015 seconds for the discrete outcome, while the QTE estimators take 0.148 and 0.145 seconds, respectively. This corresponds to our method being approximately 18 times faster for the continuous outcome and 9 times faster for the discrete outcome.

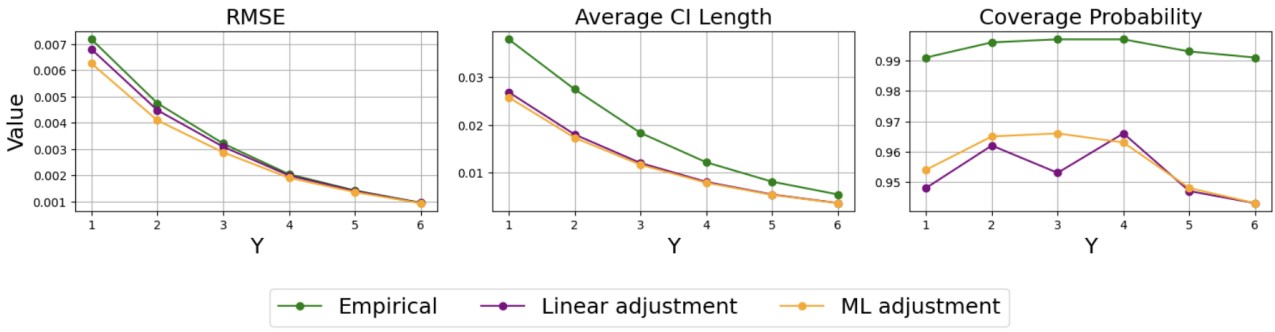

*Figure 5.* RMSE, average length and coverage probability of 95% confidence intervals (CI) on simulated data with discrete outcomes ($n = 1,000$). Linear adjustment uses linear regression, and ML adjustment uses gradient boosting, both with 2-fold cross-fitting. Number of simulations is 1,000.

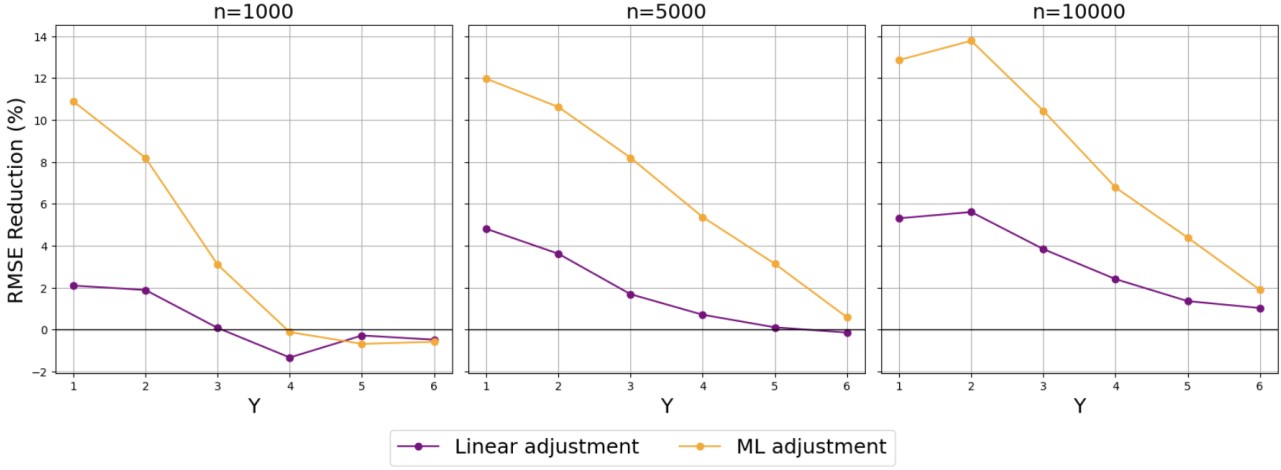

*Figure 6.* RMSE reduction (%) of regression-adjusted estimators vs. empirical estimator across quantiles for discrete outcomes, with $n \in \{1,000, 5,000, 10,000\}$. Linear adjustment uses linear regression; ML adjusment uses gradient boosting. Both use 2-fold cross-fitting. Number of simulations is 1,000.

*Table 2.* Comparison of RMSE reduction (%) between the proposed method and the estimator by Jiang et al. (2023), using linear adjustment on simulated data ($n = 1,000$)

| Method | Quantiles | | | | | | | | | Execution Time (SD) |
|---|---|---|---|---|---|---|---|---|---|---|
| | 0.1 | 0.2 | 0.3 | 0.4 | 0.5 | 0.6 | 0.7 | 0.8 | 0.9 | |
| *Continuous Outcome* | | | | | | | | | | |
| Jiang et al. (2023) | -0.55 | 0.44 | 2.82 | 3.90 | 6.51 | 5.66 | 2.96 | 2.89 | 4.45 | 0.1480 (0.0200) |
| Proposed Method | 6.42 | 9.26 | 11.67 | 11.51 | 10.79 | 11.41 | 12.48 | 11.58 | 9.91 | 0.00829 (0.0168) |
| *Discrete Outcome* | | | | | | | | | | |
| Jiang et al. (2023) | -1.97 | -2.20 | -2.59 | -4.93 | -2.69 | -4.95 | -5.19 | 0.08 | 0.38 | 0.1446 (0.01112) |
| Proposed Method | 5.62 | 5.36 | 2.10 | 0.86 | 2.67 | 2.41 | 5.67 | 3.75 | 6.10 | 0.0153 (0.00665) |

## D.2. The Impacts of Microfinance

**Dataset**  The dataset from the field experiment conducted by Attanasio et al. (2015) is available for download at OpenICPSR (Project 113597, Version V1).

Figure 7 highlights the five provinces in Mongolia where the experiment was conducted. The original map was sourced from https://en.wikipedia.org/wiki/Provinces_of_Mongolia and subsequently modified by the authors to display the provinces, their associated stratum indicators, and the location of Ulaanbaatar, the capital city.

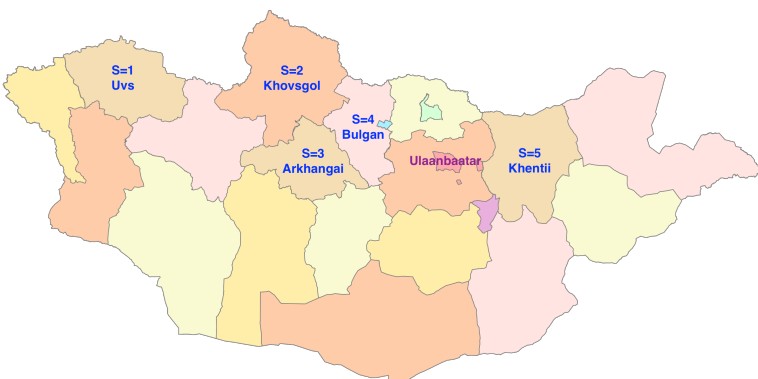

*Figure 7.* Five northern provinces in Mongolia where the experiment was conducted. Randomization was stratified at the provincial level to ensure the inclusion of all treatment groups within each province.

*Table 3.* Pre-treatment covariates included in regression adjustment

| Variable name | Description |
|---|---|
| Age | Age in years of respondent |
| Age squared | Age in years of respondent squared |
| Buddhist | Respondents is of the Buddhist religion |
| Children $< 16$ | Number of children in the household younger than 16 years |
| Education high | Dummy variable that is 1 if the respondent completed grade VIII or higher |
| Education vocational | Dummy variable that is 1 if the respondent completed vocational training |
| Female adults | Number of female household members aged 16 or older |
| Halh | Respondent ethnicity is Halh |
| Household size | Number of children and adults in the household |
| Loans at baseline | Dummy variable that is 1 if the household had at least one loan outstanding at the time of the baseline interview |
| Male adults | Number of male household members aged 16 or older |
| Married | Dummy variable that is 1 if the respondent is married or living together with partner |
| Household income | Total annual household income prior to the experiment |
| Wage income | Total annual household income from wage employment prior to the experiment |
| Enterprise revenue | Annual enterprise revenue prior to the experiment |
| Enterprise profit | Annual enterprise profit prior to the experiment |

*Table 4.* Estimated treatment assignment probabilities within each stratum (joint-liability lending vs. control). The stratum indicator $S_i$ corresponds to provinces in Mongolia, and $\widehat{\pi}(S_i)$ represents the estimated probability of assignment to the joint-liability lending.

| $S_i$ | 1 | 2 | 3 | 4 | 5 |
|---|---|---|---|---|---|
| $\widehat{\pi}(S_i)$ | 0.53 | 0.61 | 0.55 | 0.55 | 0.69 |

