# OpenReview forum: "On Efficient Estimation of Distributional Treatment Effects under Covariate-Adaptive Randomization"
_ICML.cc/2025/Conference — ICML 2025 poster_

### Official Review · Reviewer_sqwn · 2025-02-25

**Overall Recommendation:** 4

**Summary:**

This paper proposes a method of estimating the distributional treatment effect, in a setting that uses randomised experiments using covariate-adaptive randomisation. The distribution is captured through the cumulative distribution function, and estimation is done via regression-adjustment.

**Claims And Evidence:**

Theoretically, it is claimed that the proposed estimator for the distributional treatment effect is asymptotically a Gaussian process (Theorem 5.2), and that the estimator attains the semiparametric efficiency bound. I went through some of the proofs and could not find any obvious errors, and these results are very believable. Practical advantages are demonstrated through experiments.

**Essential References Not Discussed:**

Nothing essential missing I would say.

**Experimental Designs Or Analyses:**

No issues I could find.

**Methods And Evaluation Criteria:**

The method given in algorithm 1 makes sense for the problem at hand.

But I personally do not believe that the cumulative distribution function is the best way of capturing the features of a distribution, because one needs to regress for many $y$ values to get a sense of the distributional treatment effect. Quantiles, kernel mean embeddings, or even specific distributional quantities like the variance would be more of interest in my opinion. Please correct me if I'm mistaken about this, but I think the authors should at least give a discussion of this issue in the paper. See also "Questions".

**Other Comments Or Suggestions:**

216L: The indicator function is defined here, but it is actually used first on page 3. If you want to explicitly introduce the indicator function, I think it should be done when it is first used. But I think it is not strictly necessary, the vast majority of readers should be familiar with the indicator function. Indeed, I thought nothing of it when it was used on page 3 without being explicitly introduced.

Algorithm 1: This is being pedantic, but you use $w$ for both a particular treatment and as indexing through the two treatments $\{w,w'\}$. I think it would be better to use another indexing letter, what do you think?

270L: The notation $l^\infty(\mathcal{Y})$ (or $l_\infty(\mathcal{Y})$) is usually reserved for the space of bounded sequences, i.e. bounded functions on the domain $\mathbb{N}$. The notation $L_\infty(\mathcal{Y})$ would be better for the space of bounded functions with the supremum norm.

255R: In Theorem 5.2, "for uniformly over $y\in\mathcal{Y}$" sounds strange. I think "for" can be removed.

260R: I think it would be much better to be specific about what $\mathcal{G}(y)$ is here, rather than referring the reader to the appendix, especially given that this is one of the main results of the paper. The same applies for Theorem 5.3.

The reference [Bickel et al., 1993] is strange - some authors are repeated.

673: In the definition of $\mu_w(y,s)$, the closing round bracket is in the wrong place.

675: The notation $I_x(y)$ was already used on 226R, for a certain set of indices. It would be better to use another notation.

**Other Strengths And Weaknesses:**

The paper is well-written, well-structured, and delivers its messages in a clear way. It was a pleasure to read.

However, the mathematical notations and presentation could be improved somewhat, especially in Section 5. See "Other Comments Or Suggestions".

In general, I would say that this is a solid paper, but not groundbreaking. The main contribution is the proposal of the estimator itself, and the algorithm to compute it, but distributional treatment effects in the case of CAR seems already done in the form of quantile treatment effects by (Jiang et al., 2023), and to tackle the same problem with quantiles replaced by cumulative probability distributions, seems somewhat incremental. However, I don't like to see papers based on reviews of the form "incremental" or "lack of novelty", and the submission does do quite a solid job of the problem at hand.

The theory is nice to have and it is well written (with some caveats - see other sections) but there are no novel insights that arise from it - the results and proofs are very similar to those of existing papers. I think it would have been better to provide slightly novel forms of results, perhaps non-asymptotic high-probability bounds or lower bounds, but then that would go towards making this paper a theory paper, which of course this submission is not. The experiments also seem to be carried out thoroughly, but I'm not convinced that they demonstrate a clear need or advantage for this method.

**Questions For Authors:**

You use the cumulative distribution function to represent a distribution, and of course this is one good way, as it completely characterises a real distribution. However, when the outcome variable is not real-valued, or when you want to capture more specific distributional aspects, there are other ways to characterise a distribution, for example, kernel mean embeddings or kernel witness functions, as done in [Park et al., 2021], or the concurrent paper by Näf and Sussman (https://arxiv.org/abs/2411.08778, which would be relevant to cite). Do you think it would be possible to apply the kernel representations of distributions to CAR, or equivalently, to use your approach for CAR using kernel representations of distributions?

In Assumption 5.1(iii), does the constant depend on $w$ and $y_1,y_2$?

**Relation To Broader Scientific Literature:**

The authors do a great job of reviewing the related literature in Section 2, going much further back than what is usual, with papers cited from the 70s. The review of recent literature is extensive and thorough.

**Theoretical Claims:**

I checked some parts of the proofs of Theorems 5.2 and 5.3, I did not go through every line, but I could not spot any obvious errors. The proofs are essentially lifted from previous works and I can fully believe that they are true. The mathematical presentation is in general very nice, and I appreciate the effort that the authors put into writing the proofs.

However, I think the presentation of the proofs can be improved. The authors use many concepts that are taken directly from the proofs of previous papers, such as VC-type function classes or Lemma N.2 of (Jiang et al., 2023). VC-type function classes were introduced by (Chernozhukov et al., 2014) but I would not say the definition and the ensuing results are standard enough in the literature for most readers to be familiar, and it would certainly be worth giving the definitions, at least in the Appendix before the proofs start.

Also, the quantitative statements of both Theorems 5.2 and 5.3 are deferred to the Appendix and it's quite hard to find it hidden in the proofs, and I think the precise statements should be given in the theorem statements in the main body.

---

> ### Author Rebuttal · Authors · 2025-04-01
>
> We are deeply grateful for your detailed and thoughtful review and appreciate your positive assessment of our paper.
>
> **Theoretical Presentation**
>
> Thank you for the valuable suggestions to improve our mathematical presentation:
>
> 1. We will provide definitions of VC-type function classes in the Appendix as you suggested.
> 2. We will move the quantitative statements of Theorems 5.2 and 5.3 to the main body.
> 3. We will correct the notational inconsistencies you kindly pointed out, including:
>     - Moving the definition of the indicator function to its first use
>     - Using different indexing in Algorithm 1 to avoid confusion
>     - Using L∞(Y) instead of l∞(Y) for the space of bounded functions
>     - Correcting the statement "for uniformly over y∈Y" by removing "for"
>     - Fixing the bracket placement in μw(y,s) and the repeated notation of Ix(y)
>     - Correcting the Bickel et al. reference
>
> **Novelty and Contributions**
>
> Regarding your comment about incrementality compared to Jiang et al. (2023):
>
> 1. While Jiang et al. focus on quantile treatment effects under CAR, our distributional approach offers important advantages. Quantile estimation requires much stronger conditions (continuity and positive density around quantiles of interest), while our approach is applicable to non-continuous outcomes.
>
>     We add the following comparison of our method with Jiang et al’s below. We use the same DGP (our original DGP) with n = 5000 to compute the QTE. The table below presents the RMSE reduction (%) of regression-adjusted QTE estimators relative to the unadjusted QTE, following Jiang et al.'s method. Linear adjustment achieves a comparable magnitude of RMSE reduction to the DTE estimator. In contrast, logistic regression fails to reduce RMSE due to bias. The DGP in this setting is highly nonlinear and complex, with many irrelevant covariates. Consequently, simple linear adjustment outperforms ML-based adjustment in this specific case for the QTE.
>
>     | Quantiles | 0.1 | 0.2 | 0.3 | 0.4 | 0.5 | 0.6 | 0.7 | 0.8 | 0.9 |
>     | --- | --- | --- | --- | --- | --- | --- | --- | --- | --- |
>     | Jiang et al Linear (QTE) | -0.55 | 0.44 | 2.82 | 3.90 | 6.51 | 5.66 | 2.96 | 2.89 | 4.45 |
>     | Jiang et al Logistic (QTE) | -322.58 | -352.95 | -260.41 | -114.72 | -11.48 | -120.13 | -250.86 | -289.59 | -194.00 |
>
> 1. The formulation of our approach leads to a more straightforward algorithm for estimating distributional or quantile treatment effects. This offers more than mere computational advantages and delivers substantial practical improvements compared to existing methods. Please refer to the response to Reviewer G1Ux for detailed comparisons.
>
> We will clarify these distinctions more explicitly in the revised manuscript.
>
> **Regarding Your Specific Question**
>
> In terms of Assumption 5.1(iii), we realize that this assumption is more restrictive than necessary. We will revise the condition to hold locally. To prove our main results, we need the following condition:
> $\sup_{f \in \mathcal{F}}\mathbb{E}f^2 \leq c\epsilon \equiv \sigma^2$
> for some constant $c>0$, where $\mathcal{F} = \\{ \phi_w(y_2,s,Y_i^s(w),X_i^s)- \phi_w(y_1,s,Y_i^s(w),X_i^s): y_1,y_2 \in \mathcal Y, y_1 < y_2 < y_1+\epsilon \\}$ for some sufficiently small $\epsilon >0$.
>
> **Alternative Distributional Representations**
>
> Thank you for raising an inspiring question about the applicability of Kernel Mean Embedding (KME) approaches. Our simple answer regarding whether KME is applicable for generic outcome distributions is positive. For multivariate outcomes or non-standard outcomes, estimating the conditional distribution function would be challenging, while working in the Reproducing Kernel Hilbert Space (RKHS) would be more straightforward. One caveat is that if the kernel function in use is continuous, the approach requires the continuity of underlying distributions. Since your question is both challenging and important, we will include it in the discussion section as a promising avenue for future research.
>
> Thank you again for your careful reading and constructive suggestions, which will help us improve the clarity and presentation of our work.

---

> > ### Comment · Reviewer_sqwn · 2025-04-03
> >
> > Dear authors,
> >
> > Thank you for your detailed rebuttal, and I raise the score in light of it.
> >
> > Best,
> > reviewer

---

### Official Review · Reviewer_CJRK · 2025-03-14

**Overall Recommendation:** 2

**Summary:**

The authors propose an estimator and inference method based on asymptotic normality for distributional treatment effects under a covariate-adaptive randomization. The primary estimand considered in this work is the difference between cumulative distribution functions  for a fixed value $y$ between treatments. The authors propose an estimation approach based on inverse probability weighing and regression adjustment, and show that it satisfies asymptotic normality, enabling inference. Furthermore, they provide results showing semi parametric efficiency of their estimator, and provide experiments using their approach.

**Claims And Evidence:**

The claims made by the author in their theorems are well justified, and appear to follow from standard techniques in the literature.

The authors make the following claim about regression adjustment in their Introduction (l. 61): "our work advances this framework to accommodate distributional treatment effects." However, regression adjusted / doubly robust approaches to estimation of distributional causal parameters have been done extensively in previous works (including for discrete outcomes) by references the authors themselves cite (e.g. Kallus et. al., 2024, Kallus & Oprescu, 2023). These papers even extend their estimand to general distribution dissimilarity measures (such as f-divergences) between the potential outcomes of two distributions.

Could the authors elaborate a bit more on how their work positions itself uniquely beyond existing approaches?

**Essential References Not Discussed:**

The paper cites all related work in the literature, but fails to distinguish its results from the previous works. It could be helpful to contrast their results (both in terms of estimator and efficiency bound) in order to make their contributions more clear.

**Experimental Designs Or Analyses:**

N/A

**Methods And Evaluation Criteria:**

The proposed methods are tested on both empirical and real-world data. It would be helpful to have error bars in Table 1 in order to determine whether the changes are significant or not. Additionally, given that this paper provides asymptotic normality results, it would be nice to see additional experiment results that show approximate Type I Error control if using Wald-style confidence intervals for inference. .

**Other Comments Or Suggestions:**

N/A

**Other Strengths And Weaknesses:**

Strengths:
* The writing and explanation of this work is clear. For example, Assumption 3.1 is well described by the authors, with each component explained clearly.
* The authors incorporate covariate-adaptive randomization (which differs from standard context-dependent propensities slightly by assuming it only depends on the stratum).

Weaknesses:
* Unclear if these results provide additional benefits from existing work on doubly-robust estimation of distributional treatment effects
* Does not empirically validate important results in the paper, such as asymptotic normality and Type I error of its Wald confidence interval.
* Does not clearly separate its results from existing, well-established frameworks from related literature.

**Questions For Authors:**

It would be nice for the authors to clarify their contributions beyond defining an estimator for the difference in probability for outcomes under some threshold $y$ between treatments, and using the well-known doubly-robust framework for estimating this quantity. This quantity is easier to estimate than other distributional quantities, such as differences between quantiles of outcomes under each treatment, due to having no nuisances and a nice linear form. The proposed estimator appears to be a simple extension of the doubly-robust framework for causal estimates, and I'm unsure where the covariate-adaptive randomization plays a novel role in either the estimator or its analysis.

Could the authors please clarify this?

**Relation To Broader Scientific Literature:**

This paper provides an application of regression adjustment / doubly-robust approaches for covariate-adaptive randomization. The main contribution of this paper is its choice of estimand, which can be used to determine the difference in probability masses between two treatments, given upper/lower bounds on the outcome value.

The estimand and its efficiency results seem to follow directly from many existing works.

**Theoretical Claims:**

The proofs seem reasonable - I have not checked carefully.

---

> ### Author Rebuttal · Authors · 2025-04-01
>
> We sincerely appreciate your detailed review of our manuscript. Your insightful feedback has greatly helped us identify key areas to strengthen in our work.
>
> **Distinguishing Our Work from Existing Literature**
>
> You kindly raised a question about positioning our work uniquely beyond existing approaches. We acknowledge that our current manuscript does not sufficiently differentiate our contributions, and we will address this limitation in revision as follows:
>
> 1. While doubly-robust approaches for distributional treatment effects exist in the general causal inference literature including Kallus & Oprescu (2023), our work addresses the unique challenges of covariate-adaptive randomization (CAR) designs. CAR introduces specific dependence structures that require different theoretical handling than standard randomized experiments. Also, the focus of Kallus & Oprescu (2023) is the conditional distributional treatment effect rather than the unconditional distributional treatment effect that we consider.
> 2. A key distinction is that our approach directly estimates distributional effects rather than quantile treatment effects (QTE). Existing literature, including Kallus et al. (2024) and Jiang et al. (2023), focuses primarily on QTE, which requires much more stringent conditions such as smooth density around the quantiles of interest. Our approach does not require these requirements, making it applicable to a broader range of outcome distributions, including discrete and mixed distributions.
> 3. We will also add simulation results to compare our method with Jiang et al. (2023). Please see responses to Reviewer sqwn and Reviewer G1Ux.
>
> **Contributions of Our Work**
>
> 1. Our efficiency bound results are derived specifically for the CAR setting, which has not been previously established for distributional treatment effects. This distinction is important as it provides experimenters with guarantees on the maximum precision achievable in these increasingly common experimental designs.
> 2. The efficiency bound and asymptotic results under CAR differs from that under complete randomization, particularly in how the stratum indicators interact with additional covariates. Our theoretical results characterize this relationship precisely and provide a valid inferential procedure.
> 3. We think that the linear form of our estimator is a key strength compared to quantile treatment effects. This simplicity should be taken as advantageous rather than a limitation. The CAR setting introduces specific challenges that our method effectively addresses, resulting in superior practical performance compared to quantile treatment effects, which require non-differentiable optimization methods. Our approach is theoretically valid and practically useful.
>
> **Empirical Validation**
>
> Following your advice, You correctly point out limitations in our empirical validation:
>
> 1. We will add error bars to Table 1 to determine statistical significance of the improvements.
> 2. We will conduct additional experiments to validate the Type I Error control using Wald-style confidence intervals, demonstrating the practical implications of our asymptotic normality results.
>
>     We report the average length and coverage probabilities of the 95% confidence intervals based on analytic standard errors derived from our asymptotic variance, for our original DGP with n=5000. Both the linear and ML adjustment methods (using XGBoost) result in shorter confidence intervals. The coverage probabilities remain close to the nominal level of 0.95 for both the simple and adjusted estimators, with a slight over-coverage observed for certain quantiles, reaching approximately 0.97.
>
>
>     | Quantiles | Simple (CI Length) | Linear (CI Length) | XGBoost (CI Length) | Simple (Coverage) | Linear (Coverage) | XGBoost (Coverage) |
>     | --- | --- | --- | --- | --- | --- | --- |
>     | 0.1 | 0.0337 | 0.0332 | 0.0328 | 0.945 | 0.964 | 0.966 |
>     | 0.2 | 0.0456 | 0.0445 | 0.0438 | 0.964 | 0.967 | 0.970 |
>     | 0.3 | 0.0534 | 0.0517 | 0.0506 | 0.948 | 0.946 | 0.947 |
>     | 0.4 | 0.0586 | 0.0564 | 0.0548 | 0.954 | 0.955 | 0.956 |
>     | 0.5 | 0.0619 | 0.0591 | 0.0572 | 0.951 | 0.960 | 0.965 |
>     | 0.6 | 0.0637 | 0.0603 | 0.0581 | 0.963 | 0.963 | 0.964 |
>     | 0.7 | 0.0639 | 0.0598 | 0.0576 | 0.946 | 0.962 | 0.972 |
>     | 0.8 | 0.0627 | 0.0578 | 0.0558 | 0.949 | 0.958 | 0.975 |
>     | 0.9 | 0.0600 | 0.0541 | 0.0528 | 0.935 | 0.952 | 0.978 |
>
> We believe these revisions will address your concerns about the novelty and positioning of our work in relation to existing literature. Thank you again for your thoughtful feedback.

---

### Official Review · Reviewer_G1Ux · 2025-03-15

**Overall Recommendation:** 3

**Summary:**

The paper proposes a method to estimate distributional treatment effects in randomized experiments that leverages additional covariates, beyond stratum indicators, to improve precision. The authors posit a regression adjustment based on Neyman-orthogonal moment conditions to flexibly estimate the nuissance parameters using off the shelf machine learning techniques. Theoretically the paper provides asymptotic guarantees for their estimator and show that it achieves the semiparametric efficiency bound. Empirically, the paper shows through simulations that the ML adjustment reduces the variance of the estimated quantile treatment effects and the practicality of the method is shown in an application to the impacts of Microfinance.

**Claims And Evidence:**

The paper makes three main claims:

1. That it provides a new method to use auxiliary covariates in CAR to estimate quantile/distributional treatment effects that is also applicable for discrete random variables.

2. That it provides an asymptotic regime under which the limit distribution of their estimator can be derived and the estimator achieves the semi-parametric efficiency bound.

3. That it demonstrates the effectiveness of the estimator in real settings.

Overall, the paper is very well written and the theory and method are clear and convincing. However, the theory and method are very similar to Jiang et al. 2023 (JOE). The authors recognize this and suggest that their advantage lies in that their method is applicable to discrete random variables. However, the paper does not highlight this either in the theoretical section nor in the simulations. It would be beneficial to flesh out the differences with Jiang et al. 2023 and provide an example of when their method works but not Jiang's and a comparison in the simulation section.

**Essential References Not Discussed:**

The literature review is good. More comparison to Jiang et al. 2023 would be beneficial.

**Experimental Designs Or Analyses:**

It would be good to have a simulation design in which the covariates are discrete and comparison to other methods (ie. Jiang et al. 2023 if applicable, and if not, say why). Also, it would be good to comment in the simulation design on whether we expect a large or small RMSE reduction, given that the ground truth is known. It might be more compelling to add Figure D.1. in the main body given that seeing the QTE is the goal of the paper.

**Methods And Evaluation Criteria:**

The proposed method and evaluation criteria are sensible.

**Other Comments Or Suggestions:**

The paper is very well written and a pleasure to read! I think it is very polished and convincing, my only caveat is its novelty with respect to Jiang et al. 2023.

**Other Strengths And Weaknesses:**

See above.

**Questions For Authors:**

See above.

**Relation To Broader Scientific Literature:**

The key contribution is to provide a new method for QTE/DTE for CAR with regression adjustments to improve the variance that also works with discrete random variables. This appears to be a useful addition to the literature but the novelty relative to Jiang et al. 2023 should be better framed.

**Theoretical Claims:**

The theoretical derivations and statements are polished and appear correct, but both the setting, proofs and theoretical statement seem to rely on Jiang et al. 2023. It would be beneficial to discuss the differences more and where the assumptions of each paper differ. For example, Assumption 5 (ii) etc.

---

> ### Author Rebuttal · Authors · 2025-04-01
>
> We are grateful for your thoughtful and constructive feedback on our paper. Your positive assessment regarding clarity and theoretical development is encouraging.
>
> Following your comments, we will highlight three key distinctions from Jiang et al. (2023) in the revision:
>
> 1. **Discrete Random Variables**: Our method is applicable for any type of outcome variables, including discrete or mixed-type outcome variables that are common in real-world applications. In the revision, we will add a simulation design featuring discrete outcomes to demonstrate this advantage.
> 2. **Computational Benefits**: Our approach offers significant computational advantages that were not emphasized enough in the current version. Jiang et al.'s algorithm minimizes a non-differentiable objective function through multi-step optimization, and their last step potentially involves a grid search within the estimated interval. Our approach is much more straightforward in practice. Our simulation study suggests that this leads to substantial efficiency gains without sacrificing accuracy.
> 3. **Relaxed Assumptions**: Our framework operates under less restrictive assumptions. In particular, our approach relaxes several continuity requirements that are present in Jiang et al. We will elaborate on these differences in Assumption 5(ii) and other relevant theoretical sections to clarify how our method maintains asymptotic guarantees under broader conditions.
>
> **Addressing Specific Recommendations**
>
> - We will revise Section 2 to carefully explain distinctions from Jiang et al. (2023).
> - We will add a new simulation with discrete outcomes and discrete covariates and direct comparison to Jiang et al. where applicable.
>
>     First, we consider the same DGP setup as before but with a discrete outcome variable following a Poisson distribution. The conditional mean is given by
>
>     $E[Y_i | X_i, W_i, Z_i] = 0.2 \left| b(X_i) + c(X_i)W_i + \gamma Z_i + u_i \right|.$
>
>     The table below presents the RMSE reduction (%) of our proposed DTE estimators relative to unadjusted estimators with n=1000. Both linear and logistic regression achieve reductions ranging from -1% to 6%. Under this DGP, computing QTE is not meaningful, as most quantiles equal zero due to a large mass point at zero.
>
>     | Location | 0 | 1 | 2 | 3 | 4 | 5 | 6 | 7 | 8 | 9 | Execution time |
>     | --- | --- | --- | --- | --- | --- | --- | --- | --- | --- | --- | --- |
>     | Proposed Linear (DTE) | 1.29 | 2.10 | 1.89 | 0.08 | -1.33 | -0.28 | -0.48 | -0.61 | 0.28 | -0.22 | 0.0120s (SD 0.00632) |
>     | Proposed Logistic (DTE) | 2.49 | 5.81 | 4.59 | 2.38 | 0.71 | -0.45 | -0.44 | -0.13 | 0.25 | -0.28 | 0.0246s (SD 0.0144) |
>
>     Next, we consider a setting with discrete covariates. Under the same DGP setup, we sample X independently from Uniform(−5,5)and round each value to the nearest integer. In this case, the DTE estimators achieve RMSE reductions ranging from 0.8% to 6.1% for linear regression and from 0.9% to 12.6% for logistic regression. However, the QTE estimator fails to yield reductions across most quantiles due to the discrete nature of the outcome variable.
>
>     In terms of computational efficiency, the DTE estimators require 0.01s for linear regression and 0.04s for logistic regression, whereas the QTE estimators take 0.14s and 0.22s, respectively. This implies that our method is approximately 10 times faster for linear regression and 6 times faster for logistic regression.
>
>     | Quantiles | 0.1 | 0.2 | 0.3 | 0.4 | 0.5 | 0.6 | 0.7 | 0.8 | 0.9 | Execution time |
>     | --- | --- | --- | --- | --- | --- | --- | --- | --- | --- | --- |
>     | Jiang Linear (QTE) | -1.97 | -2.20 | -2.59 | -4.93 | -2.69 | -4.95 | -5.19 | 0.08 | 0.38 | 0.1446s (SD 0.0111) |
>     | Jiang Logistic (QTE) | -123.03 | -313.59 | -381.79 | -251.43 | -22.60 | -206.06 | -454.47 | -171.93 | -115.41 | 0.2206s (SD 0.0164) |
>     | Proposed Linear (DTE) | 5.62 | 5.36 | 2.10 | 0.86 | 2.67 | 2.41 | 5.67 | 3.75 | 6.10 | 0.0153s (SD 0.0067) |
>     | Proposed Logistic (DTE) | 2.38 | 3.12 | 0.97 | 0.51 | 2.19 | 3.28 | 7.05 | 8.97 | 12.62 | 0.0386s (SD 0.0106) |
> - Figure D.1 will be moved to the main body to better illustrate the quantile treatment effects, while emphasizing that our main focus is on DTE or PTE rather than QTE.
> - We will enhance our simulation discussion with expected RMSE reduction context given the known ground truth, , when possible. In most cases, however, we have chosen relatively complex processes where theoretical expectations may not be available.
>
> These revisions will clarify the novelty and practical advantages of our approach while maintaining the strong theoretical foundation you recognized in your review.
>
> We appreciate your comments and would be happy to elaborate further on any questions or concerns you might have.

---

> > ### Comment · Reviewer_G1Ux · 2025-04-03
> >
> > Thank you for your responses. I think the key point for the paper to be accepted is whether it gives a theoretical advantage over Jiang et al. 2023. I appreciate the discussion and additional simulations. If the other referees agree that change in Assumption 5.1 is reasonable in allowing discrete covariates, I will raise my score to 4.

---

> > > ### Author Response · Authors · 2025-04-08
> > >
> > > Thank you very much for your feedback. We are pleased to highlight that both Reviewer qJAh and Reviewer sqwn, who previously expressed concerns regarding Assumption 5.1, have subsequently raised their scores. We greatly appreciate your comments, which have helped us improve the clarity and strength of the paper.

---

### Official Review · Reviewer_qJAh · 2025-03-21

**Overall Recommendation:** 3

**Summary:**

The paper develops a regression‐adjusted estimator for distributional treatment effects (DTEs) under covariate‐adaptive randomization (CAR). It presents (i) a derivation of the estimator’s limit distribution under CA, (ii) a semiparametric efficiency bound result, and (iii) empirical/simulation demonstrations (including a microcredit dataset). The main claim is that incorporating extra covariates (beyond basic stratum indicators) can reduce variance in estimating how an intervention shifts the entire outcome distribution.

## update after rebuttal

The authors have addressed some of my concerns regarding discrete outcomes and finite sample performance. As a result, I have raised my score from **2** (Weak Reject) to **3** (Weak Accept). I still find the contribution somewhat incremental, and believe the work could be further strengthened with clearer theoretical development and a discussion of computational limitations. Thus, I maintain a borderline positive score to be considered alongside the evaluations of the other reviewers.

**Claims And Evidence:**

* Claims:
	* The proposed estimator is unbiased and consistent for the DTE.
	* Adjusting for covariates improves precision under CAR.
	* The proposed estimator attains the semiparametric efficiency bound.
* Evidence:
	* They provide formal asymptotic proofs for consistency and efficiency.
	* Simulation and one real-data example give some empirical support.
	* The real-data example has a small sample, so results there are less conclusive.

Overall, the claims seem well supported. However, the paper should have a more detailed discussion (and theoretical results) of performance in finite samples that is common with doubly robust/Neyman‐orthogonal style estimators. In addition, although the introduction asserts that the method can handle discrete outcomes, Assumption 5.1 requires $\mu_w(y, S, X)$ to be continuous in $y$, which does not apply for discrete outcomes. Thus, there appears to be a mismatch between the paper’s stated applicability and the continuity assumption in its theoretical results.

**Essential References Not Discussed:**

The paper might profit from a deeper discussion of Kallus, Mao, & Uehara (2019) regarding more efficient approaches for quantile (and possibly distributional) estimation approaches.

**Experimental Designs Or Analyses:**

* The simulation design is reasonably standard for demonstrating coverage and bias. However, the gains in RMSE are moderate, so real‐world improvement may be small.
* Since the synthetic experiments are effectively looking at quantiles, the authors should compare with  Jiang et al. 2023 as a baseline.
* The real microcredit experiment is interesting but limited by sample size, reducing the credibility of the precision‐gain claims.

**Methods And Evaluation Criteria:**

* The proposed method logically follows from doubly robust/Neyman‐orthogonal style estimators, adapted to CAR for DTEs.
* Using standard approaches (like cross‐fitting and machine‐learning regression) is sensible, although carrying it out “for every $y\in \mathcal{Y}$” can be computationally expensive for continuous $y$ (Algorithm 1 in the paper). The authors should consider [1] or similar since their method may be more computationally tractable for handling continuous $y$ (although [1] only applied to quantile effects)

[1] Kallus, N., Mao, X., & Uehara, M. (2019). Localized debiased machine learning: Efficient inference on quantile treatment effects and beyond. arXiv preprint arXiv:1912.12945.

**Other Comments Or Suggestions:**

**General comment:** The paper is neither here nor there. It needs more theoretical results for finite samples to justify the robustness/orthogonality results, it needs an alternative method for estimating with continuous $Y$ (otherwise, one can just use the quantile paper of  Jiang et al. (2023)), and it feels incremental in the sense of “just adding DR/Neyman-orthogonal logic” to existing estimators. Furthermore, I don't think the current approach  properly handles discrete $Y$. Thus, I lean towards rejection, but I am open to reconsidering my rating.

**Other Strengths And Weaknesses:**

To summarize from the scattered sections above:

* Strengths
	* The paper applies doubly‐robust methodology to distributional treatment effect (DTE) estimation under covariate‐adaptive randomization (CAR).
	* It gives a rigorous semiparametric efficiency analysis, aligning with modern causal‐inference techniques.
	* The method is illustrated both in simulations and a real‐world microcredit dataset.
* Weaknesses
	* Some incremental feel: the main novelty is “standard DR logic” extended from average effects to distributional effects under CAR.
	* Computational impracticalities for “every $y$” are not fully addressed.
	* Gains in finite samples (especially in the real dataset) are small, so the practicality is unclear. Also, should add Jiang et al. (2023) as benchmark for the synthetic experiments.
	* The paper claims the ability to handle discrete outcomes; however, Assumption 5.1 requires $\mu_w(y, S, X)$ to be continuous in $y$, which does not align with discrete $Y$ distributions. This mismatch undermines the stated applicability and advantage over, for instance, Jiang et al. (2023).

**Questions For Authors:**

See concerns listed in the "Weaknesses" section above.

**Relation To Broader Scientific Literature:**

* This work extends prior analyses of regression adjustment under CAR from average to distributional effects (cf. Rafi 2023, Jiang et al. 2023).
* The closest relevant approach is  Jiang et al. (2023) which studies quantile treatment effects under CAR
* A mention of specialized methods for quantile treatment effect estimation (e.g., Kallus et al., 2019) would be important—especially since their approach may be more computationally tractable than doing a separate regression for each $y$.

**Theoretical Claims:**

* The authors prove asymptotic normality and derive a semiparametric efficiency bound. I skimmed the proofs and they appear sound, but Assumption 5.1 (especially (i)) might not be standard in the semi-parametric efficiency literature and require more discussion.
* They do not provide much discussion about finite‐sample guarantees besides simulations.

---

> ### Author Rebuttal · Authors · 2025-04-01
>
> We are grateful to your detailed review of our manuscript with constructive criticism, which has helped us identify important areas for improvement.
>
> **Discrete Outcomes and Assumption 5.1**
>
> You raised an important point regarding our claim about handling discrete outcomes versus Assumption 5.1 requiring continuity. This was imprecisely stated in our manuscript. To clarify:
>
> Our method allows for discrete outcomes. We will revise Assumption 5.1 to explicitly accommodate mass points in the distribution while maintaining asymptotic guarantees. The continuity assumption was unnecessarily restrictive in our presentation. More precisely, in order to prove stochastic equicontinuity for continuous or finite discrete outcomes, we need
>
> $\sup_{f \in \mathcal{F}}\mathbb{E}f^2 \leq c\epsilon$
> for some constant $c>0$, where $\mathcal{F} = \\{ \phi_w(y_2,s,Y_i^s(w),X_i^s)- \phi_w(y_1,s,Y_i^s(w),X_i^s): y_1,y_2 \in \mathcal Y, y_1 < y_2 < y_1+\epsilon \\}$ for some sufficiently small $\epsilon >0$.
>
> If discrete outcome has infinitely many points in its support, one can impose the following assumption: there exists constants $C_1, C_2>0$ such that
>
> $\sup_{f \in \mathcal{F}}\mathbb{E}f^2 \leq   c_1  \epsilon+ C_2e^{-\alpha/\epsilon}$.
>
> Also, we will add a discrete outcome simulation to demonstrate this capability, which is an advantage over some existing methods. See response to Reviewer G1Ux for simulation results with discrete outcome.
>
> **Computational Considerations**
>
> We agree that computational complexity deserves more attention.
>
> 1. Since the outcome variable of interest is one-dimensional, our algorithm can handle most problems within a reasonable timeframe. Indeed, our algorithm is more straightforward and faster than the ones in the papers you kindly raise.
> 2. Our approach offers computational advantages over quantile-based methods, because our method does not require minimizing a non-differentiable objective function through multi-step optimization and also do not rely on sorting algorithm.
> 3. We review the algorithm of Kallus et al. (2019) as you suggested and we will add some discussion on the algorithmic difference as follow:
>
>     Kallus et al. (2019) address problems in which nuisance parameters depend on the target parameter itself, as seen in cases like the quantile treatment effect (QTE) and local QTE. In contrast, our estimation of the distributional treatment effect (DTE) involves nuisance parameters that correspond to conditional means, which can be effectively estimated using machine learning algorithms.
>
>
> **Contributions**
>
> While our approach extends doubly-robust methodology to DTEs under CAR, this extension provides several valuable contributions:
>
> 1. It addresses the specific challenge of CAR designs that are increasingly common in practice but require distinct statistical treatment.
> 2. Our framework unifies treatment effect estimation across outcome types under a single methodology with less restrictive assumptions.
>
> In our revision, we will further elaborate on these contributions and highlight their practical significance.
>
> **Finite Sample Performance**
>
> We acknowledge your concern about finite sample performance:
>
> 1. We will add theoretical results on finite sample performance in our revision. As shown in our paper, the proposed estimator is unbiased in finite samples. Additionally, we can demonstrate that the variance of our estimator in finite samples is smaller than that of the standard estimator up to a deviation of order O(n^{-1}). If you are instead asking about non-asymptotic bounds, we would appreciate if you could kindly share particular papers in your mind. We are not aware of such papers and will consider this direction for future work.
> 2. The real-world gains in the microcredit example will be better contextualized by benchmarking against expected improvements based on sample size. This is usually the case where the regression-adjustment method is needed in practice. Researchers want to reduce the variance without any additional cost and our method highlights that our method with proper experimental design can significantly reduce the variance of the treatment effect estimator.
>
> **Additional Comparisons**
>
> Following your recommendations, we will:
>
> 1. Add Jiang et al. (2023) as a benchmark in our synthetic experiments — Please see responses to Reviewer sqwn and Reviewer G1Ux
> 2. Include discussion of Kallus et al. (2019)
> 3. Provide clearer comparative analysis of computational versus statistical efficiency
>
> We believe addressing these points will substantially strengthen our paper. We appreciate your thorough review and feedback.

---

> > ### Comment · Reviewer_qJAh · 2025-04-04
> >
> > Dear authors,
> >
> > Thank you for addressing my concerns. I have raised my score to reflect this.

---

### Decision · Program_Chairs · 2025-05-01

**Decision:**

Accept (poster)

**Comment:**

This paper develops a regression-adjusted estimator for distributional treatment effects under covariate-adaptive randomization. The reviewers found the paper to be theoretically sound, and their concerns were satisfactorily addressed in the rebuttal. However, the authors are encouraged to incorporate the reviewers’ suggestions into the final version.